EMBO
Molecular Medicine

# SEMA3C drives cancer growth by transactivating multiple receptor tyrosine kinases via Plexin B1

James W Peacock[1,2], Ario Takeuchi[1,3], Norihiro Hayashi[1,4], Liangliang Liu[1], Kevin J Tam[1,2], Nader Al Nakouzi[1], Nastaran Khazamipour[1], Tabitha Tombe[1], Takashi Dejima[1,3], Kevin CK Lee[1], Masaki Shiota[1,3], Daksh Thaper[1,2], Wilson CW Lee[1], Daniel HF Hui[1], Hidetoshi Kuruma[1], Larissa Ivanova[1], Parvin Yenki[1,2], Ivy ZF Jiao[1], Shahram Khosravi[1], Alice L-F Mui[5], Ladan Fazli[1], Amina Zoubeidi[1,2], Mads Daugaard[1,2], Martin E Gleave[1,2,*] iD & Christopher J Ong[1,2,**] iD

## Abstract

**Growth factor receptor tyrosine kinase (RTK) pathway activation is a key mechanism for mediating cancer growth, survival, and treatment resistance. Cognate ligands play crucial roles in autocrine or paracrine stimulation of these RTK pathways. Here, we show SEMA3C drives activation of multiple RTKs including EGFR, ErbB2, and MET in a cognate ligand-independent manner via Plexin B1. SEMA3C expression levels increase in castration-resistant prostate cancer (CRPC), where it functions to promote cancer cell growth and resistance to androgen receptor pathway inhibition. SEMA3C inhibition delays CRPC and enzalutamide-resistant progression. Plexin B1 sema domain-containing:Fc fusion proteins suppress RTK signaling and cell growth and inhibit CRPC progression of LNCaP xenografts post-castration *in vivo*. SEMA3C inhibition represents a novel therapeutic strategy for treatment of advanced prostate cancer.**

**Keywords** Semaphorin 3C; Plexin B1; prostate cancer; receptor tyrosine kinase; apoptosis

**Subject Categories** Cancer; Urogenital System; Vascular Biology & Angiogenesis

## Introduction

Androgen deprivation therapy (ADT) is first-line systemic therapy for men with metastatic prostate cancer (PCa). Unfortunately, the survival benefit from ADT is limited by emergence of lethal CRPC (Bruchovsky *et al*, 1989; Goldenberg *et al*, 1988). Development of

CRPC is a complex process that has been attributed to a variety of mechanisms including reactivation of the androgen receptor (AR) axis and activation of growth factor signaling pathways (Yap *et al*, 2011). While many growth factor receptor pathways are activated in PCa progression (such as epidermal growth factor receptor (EGFR), ErbB2 (HER2/neu), and MET), ligands that drive activation of these pathways remain poorly defined.

Our gene expression profiling data identified SEMA3C, a member of the secreted class 3 semaphorins, as a highly expressed gene in CRPC and AR pathway inhibitor-recurrent tumors. SEMA3C was prioritized as a putative CRPC driver as it was identified in two independent genomewide profiling studies. In search of novel targets associated with expression of PTEN, a gene that is frequently mutated in advanced PCa, DNA microarray profiling identified SEMA3C among the top three most differentially expressed genes between PTEN$^{+}$ vs. PTEN$^{-/-}$ cancer cells (Peacock *et al*, 2009). In addition, SEMA3C was found to be an NFκB-regulated survival factor induced by CLU (Zoubeidi *et al*, 2010b), a molecular chaperone associated with treatment resistance, suggesting SEMA3C may be functionally relevant in CRPC (Zoubeidi *et al*, 2010a). As a whole, semaphorins are a large family of evolutionarily conserved secreted or cell surface signaling proteins originally discovered as important mediators of cell migration and axon guidance in the developing nervous system (Cagnoni & Tamagnone, 2014). While semaphorins have been best characterized in the nervous system, they have also been implicated in a variety of dynamic physiological processes including angiogenesis, tissue morphogenesis, immunity, and cancer (Cagnoni & Tamagnone, 2014).

Semaphorins exhibit diverse activities in cancer. Depending on the tumor type and context, various class 3 semaphorins either promote (SEMA3C and 3E) or suppress (SEMA3A, 3B, 3D, 3E, and 3F) cancer growth (Rehman & Tamagnone, 2013). Among the class 3 semaphorins, SEMA3C is notable for its frequent association with

1 Vancouver Prostate Centre, Vancouver, BC, Canada
2 Department of Urologic Sciences, University of British Columbia, Vancouver, BC, Canada
3 Department of Urology, Graduate School of Medical Sciences, Kyushi University, Fukuoka, Japan
4 Department of Urology, The Jikei University School of Medicine, Tokyo, Japan
5 Department of Surgery, University of British Columbia, Vancouver, BC, Canada
  *Corresponding author. Tel: +1 6046 752568; E-mail: m.gleave@ubc.ca
  **Corresponding author. Tel: +1 6048 754111 ext. 63120; E-mail: chris.ong@ubc.ca or Chriso@mail.ubc.ca

tumor progression and poor prognosis across multiple tumor types including lung, breast, gastric, and ovarian cancers as well as glioblastoma (Rehman & Tamagnone, 2013). Increased expression of SEMA3C is associated with poor prognosis and tumor progression in a number of cancers (Yamada *et al*, 1997; Martin-Satue & Blanco, 1999; Konno, 2001; Galani *et al*, 2002; Herman & Meadows, 2007; Miyato *et al*, 2012; Xu *et al*, 2017). In PCa, SEMA3C promotes cell migration and invasion *in vitro* (Herman & Meadows, 2007) and drives EMT and stemness (Tam *et al*, 2017) and SEMA3C expression is a predictive marker for biochemical recurrence (BCR) (Li *et al*, 2013). Therefore, we investigated whether SEMA3C could be a key growth factor that drives CRPC progression and treatment resistance, and set out to develop a therapeutic protein inhibitor of SEMA3C signaling.

# Results

## Increased SEMA3C expression is associated with CRPC bone metastases

To define SEMA3C expression levels in benign and cancerous prostate specimens and bone metastases, we assessed the levels of SEMA3C by tissue microarray (TMA) immunohistochemical (IHC) staining of a panel of 280 human PCa specimens representing benign prostatic hyperplasia (BPH, $n = 12$), untreated hormone naive ($n = 114$), neo-adjuvant hormone therapy (NHT)-treated ($n = 87$), NHT- and docetaxel-treated ($n = 53$) radical prostatectomy PCa specimens, as well as CRPC bone metastases ($n = 30$) collected immediately after death via University of Washington Rapid Autopsy program (Rocchi *et al*, 2005). Specimens were graded on a 0–3, intensity scale representing the range from no staining to high staining by visual scoring and automated quantitative image analysis. As shown in Fig 1A and B, increased SEMA3C expression was associated with advanced PCa that have been heavily treated with NHT and docetaxel (DTXL) (\*$P = 0.02$), and CRPC bone metastases (\*\*$P = 0.0075$; Fig 1B). We validated the specificity of SEMA3C (N20) antibodies to detect SEMA3C in DU145 cells transfected with scramble (siScr) or SEMA3C siRNA (siSEMA3C-1) using confocal immunofluorescent microscopy. The siSEMA3C treatment reduced SEMA3C staining compared to siScr control cells (Appendix Fig S1A).

Next, we examined SEMA3C expression in a panel of cell lines including benign prostate epithelial cells as well as androgen-sensitive, castrate-resistant, enzalutamide-resistant, and androgen receptor-negative cell lines. Elevated secreted SEMA3C levels were found in conditioned media of castrate-resistant C4-2 and 22Rv1, enzalutamide-resistant MR49F cells, and androgen-independent DU145 cells and androgen-sensitive LNCaP cells (Fig 1C and Appendix Fig S1B) compared to two benign immortalized prostatic cells, RWPE-1 and BPH-1 cells (Appendix Fig S1B). We have observed all forms of SEMA3C secreted from all of the prostate cancer cell lines. There is typically a doublet that runs at about 83 kDa that likely represents the full-length protein and the C-terminal cleavage product often referred to as the Δ13, respectively. We also typically see a band that runs below 70 kDa that represents the p65 cleavage product. As compared to prostate cancer cell lines, secreted SEMA3C expression was very low in immortalized benign prostate epithelial cell lines, RWPE-1 and BPH-1, respectively (Appendix Fig S1B).

Mining of gene expression profiling data from Chen *et al* (2004) revealed an association between increased SEMA3C expression and resistance to anti-androgen therapy in six out of seven isogenic hormone-sensitive and castration-resistant xenograft pairs [NCBI GEO GDS535 (Barrett *et al*, 2007)] (Appendix Fig S1C). Consistent with these findings, we found a trend of higher levels of SEMA3C mRNA in castration-resistant LNCaP xenograft tumors as compared to tumors from non-castrate male mice ($P = 0.075$) (Appendix Fig S1D). We noted that a subgroup (approx. 50%) of the CRPC tumors had high SEMA3C, which suggests that SEMA3C overexpression may represent one mechanism to achieve castration resistance.

## SEMA3C activates signaling through EGFR, HER2, and MET receptor tyrosine kinases

To identify potential signaling pathways regulated by SEMA3C in LNCaP cells, we performed an unbiased proteomic screen using the Kinex KAM-1.1 antibody microarray chip from Kinexus Biosciences Corp., containing over 650 antibodies, including > 270 phospho-site-specific antibodies as well as antibodies for detection of > 240 protein kinases, 28 phosphatases, and 90 other cell signaling proteins that regulate cell proliferation, stress, and apoptosis. The levels of signaling proteins were compared in SEMA3C-overexpressing LNCaP cells to empty vector-transduced LNCaP cells, and in parallel in SEMA3C antisense knockdown compared to scrambled oligonucleotide control-treated LNCaP cells. Intriguingly, the Kinex screen revealed a number of key phosphoproteins that were upregulated by SEMA3C overexpression and correspondingly, downregulated by SEMA3C silencing (Appendix Table S1), including Erb family RTKs (ErbB2 and EGFR), PI3K/PTEN cell survival pathway proteins (Akt1 and 4EBP1), and cell cycle regulatory proteins (Rb and CDK1/2), implicating the EGFR/ErbB2 pathway in SEMA3C signaling. To investigate whether naturally secreted SEMA3C could activate the EGFR/ErbB2 signaling pathway, we treated LNCaP cells with conditioned medium (CM) harvested from HEK 293T cells that overexpress full-length wild-type SEMA3C. We observed a dosage-dependent increase in EGFR, SHC, and MAPK phosphorylation with increasing concentration of SEMA3C containing CM and a corresponding decrease in EGFR, SHC, and MAPK phosphorylation in SEMA3C immuno-depleted CM, suggesting that SEMA3C is an autocrine growth factor that drives EGFR activation (Appendix Fig S1E).

To validate whether SEMA3C activates the EGFR/ErbB2 pathway, LNCaP cells were treated with increasing concentrations of full-length recombinant SEMA3C-Fc fusion protein (SEMA3C:Fc) (0–2 μM) for 10 min and phosphorylation of EGFR, ErbB2 (HER2), and downstream signaling proteins (SRC, SHC, and MAPK) was analyzed by immunoblotting. SEMA3C treatment triggered dose-dependent increase in phosphorylation of EGFR, ErbB2, SRC, SHC, and p44/42 MAPK in LNCaP cells (Fig 1D and Appendix Fig S2A). SEMA3C stimulated EGFR and MAPK phosphorylation at levels as low as 1.6 nM SEMA3C in LNCaP cells (Appendix Fig S2B). Similarly, SEMA3C induced dose-dependent activation of EGFR signaling pathway in DU145 cells (Appendix Fig S2C).

To investigate the relationship between SEMA3C expression and EGFR/ErbB2 signaling in clinical PCa samples, we analyzed phosphorylation status of EGFR, ErbB2, and downstream signaling

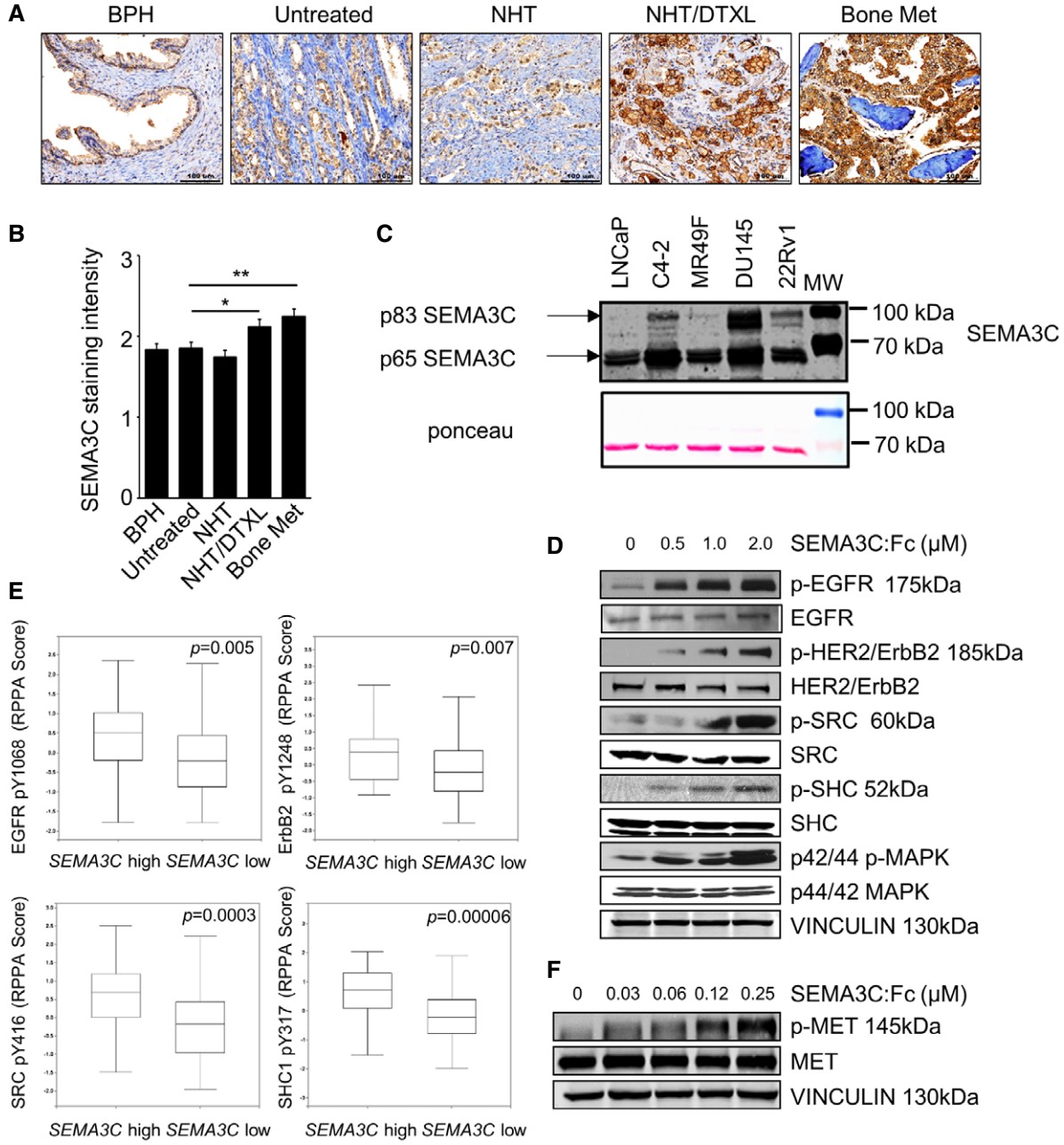

**Figure 1.  Increased SEMA3C expression correlates with CRPC and SEMA3C activates EGFR, ErbB2, MET, and SRC tyrosine kinase signaling.**

A   Representative SEMA3C immunostaining of BPH, untreated hormone naïve, neo-adjuvant hormone therapy (NHT)-treated, NHT- and docetaxel-treated radical prostatectomy PCa specimens, and a bone CRPC metastasis specimen. Scale bar: 100 µm.

B   IHC intensity scores of SEMA3C (N20) staining from BPH, untreated, NHT-treated, NHT- and docetaxel-treated, and bone metastasis. Mean ± SEM; *P = 0.02, **P = 0.0075. Statistical analysis was performed using the unpaired two-tailed Student's t-test.

C   Immunoblot analyses of SEMA3C levels in conditioned media produced from androgen dependent (LNCaP), castration-resistant (C4-2, 22Rv1), enzalutamide-resistant (MR49F), and androgen receptor-negative (DU145) PCa cells seeded at equivalent cell density. Whole-cell lysates were immunoblotted with Ponceau staining as control. The data are representative of three independent experiments.

D   Activation of EGFR, HER2/ErbB2, and downstream signaling in LNCaP cells treated with varying concentrations of SEMA3C:Fc (0–2 µM) for 10 min. Levels of indicated phosphoproteins and total proteins were assessed by immunoblot analyses. Vinculin is shown as loading control.

E   Boxplots of RPPA measurements of indicated phosphoprotein levels in 498 patient tumor samples expressing SEMA3C high versus SEMA3C low mRNA levels from TCGA prostate adenocarcinoma provisional data set. Boxes span the interquartile range. Horizontal line within the box represents the median phosphoprotein levels. Error bars represent the range from the highest to the lowest observations. Statistical analysis was performed using the unpaired two-tailed Student's t-test.

F   SEMA3C activates MET in a dose-dependent manner. DU145 cells were serum-starved and then stimulated with SEMA3C at the indicated doses for 10 min. Activation of MET was determined by immunoblotting with phospho-MET Abs.

Source data are available online for this figure.

proteins SRC and SHC in high SEMA3C versus low SEMA3C-expressing samples from The Cancer Genome Atlas (TCGA, http://cancergenome.nih.gov/) prostate adenocarcinoma data set using cBioPortal tools (Cerami *et al*, 2012; Gao *et al*, 2013). As shown in Fig 1E, high SEMA3C mRNA levels were associated with increased levels of phospho-EGFR (Y1068), phospho-ErbB2 (Y1248), phospho-SRC (Y416), and phospho-SHC (Y317) as determined by reverse-phase protein array (RPPA) measurements of phosphoprotein levels in patient tumor samples expressing increased *SEMA3C* mRNA with z-score threshold >1.0 (*SEMA3C* high) versus unaltered SEMA3C (*SEMA3C* low) tumor samples from TCGA prostate adenocarcinoma provisional data set. Plot and *P* value were generated from cBioPortal.

Semaphorin receptors such as Plexin B1 are also known to mediate signaling through MET receptor tyrosine kinase (RTK) (Giordano *et al*, 2002). To examine whether SEMA3C can trigger MET signaling, MET-expressing DU145 cells were stimulated with varying concentrations of SEMA3C:Fc and activation of MET signaling was examined by immunoblotting for phospho-MET. MET phosphorylation was induced by SEMA3C stimulation in DU145 in a dose-dependent manner (Fig 1F).

To determine whether SEMA3C can drive activation of EGFR and MET signaling in other cell types, we first screened cBioPortal for Cancer Genomics data for tumor types that showed an association of high SEMA3C expression and poor prognosis. High SEMA3C expression was associated with poor overall survival in renal clear cell carcinoma and bladder carcinoma (Appendix Fig S2D). Furthermore, SEMA3C has recently been shown to promote tumorigenicity and survival of glioma stem cells (Man *et al*, 2014). Hence, we screened a panel of cells lines representing renal cancer (CAKI-1, CAKI-2, ACHN), bladder cancer (UC13, T24), and glioblastoma (U87MG), for expression of SEMA3C, EGFR, MET, Plexin B1, Plexin D1, NRP1, and NRP2 (Appendix Fig S2E). Treatment of CAKI-2, T24, and U87MG cells which express lower SEMA3C levels with recombinant SEMA3C:Fc showed activation of MET and EGFR signaling in a dose-dependent manner (Appendix Fig S2F). Conversely, siRNA-mediated silencing of SEMA3C reduced levels of phospho-EGFR proteins in high SEMA3C-expressing CAKI-1 and ACHN cell lines (Appendix Fig S2G). These data collectively suggest that

SEMA3C may be a key driver of EGFR and MET signaling in a broad spectrum of cancers.

## SEMA3C drives signaling and growth via Plexin B1 and NRP1/2

Plexins and neuropilins are cell surface receptors responsible for secreted class 3 semaphorin signal transduction (Rehman & Tamagnone, 2013). Since Plexin B1 and D1 are known to mediate RTK activation such as HER2/ErbB2 (Swiercz *et al*, 2004, 2008; Casazza *et al*, 2010) and MET (Giordano *et al*, 2002), we therefore sought to examine whether NRP1, NRP2, Plexin B1, and Plexin D1 are receptors for SEMA3C in PCa cells. To this end, we first performed proliferation assays in DU145 cells treated with siRNA targeting Plexin D1 (siPLXND1), Plexin B1 (siPLXNB1), NRP1 (siNRP1), and NRP2 (siNRP2) (Fig 2A and Appendix Fig S2H). Knockdown of Plexin B1, NRP1, NRP2, and both NRP1 and NRP2 inhibited SEMA3C-induced growth of DU145 cells compared to siScramble (siScr) controls, whereas siRNA-mediated silencing of Plexin D1 did not (Fig 2A). Moreover, knockdown of Plexin B1 and of NRP1 and NRP2 individually and combined inhibited SEMA3C-activated phosphorylation of EGFR, HER2/ErbB2, and SHC compared to siScr control (Appendix Fig S2I–K). Taken together, these data suggest that Plexin B1 and coreceptors, NRP1 and NRP2, mediate SEMA3C signaling in prostate cancer cells.

The proximity ligation assay (PLA) is a powerful technique used to visualize and quantitate protein–protein interactions (see Appendix Materials and Methods). The assay reports proximal association of two target proteins less than 40 nm apart (Fredriksson *et al*, 2002). Using PLA, we sought to determine whether SEMA3C treatment of DU145 cells could induce formation of a ligand–receptor complex comprised of SEMA3C with NRP1 and Plexin B1 on the cell surface and whether Plexin B1 associates with EGFR, HER2, and MET, thereby mediating transduction of downstream RTK signaling. As a first step, we used PLA to characterize the formation of ligand–receptor complexes between SEMA3C and its putative candidate receptors, Plexin B1 and NRP1 in siScr or siPLXNB1-treated DU145 cells stimulated with either recombinant full-length SEMA3C-Fc fusion (SEMA3C:Fc) or human IgGFc (rhIgGFc) as control. We found significant direct binding of SEMA3C:Fc with Plexin B1 (Fig 2B)

---

**Figure 2. SEMA3C acts through Plexin B1.**

A    Proliferation assay of DU145 cells treated with scramble (siScr) or siRNAs specific for PLXND1, PLXNB1, NRP1, NRP2, and NRP1 and NRP2 together. SEMA3C-induced growth is expressed as percent of control (siScramble) using the crystal violet proliferation assay. Mean ± SEM; *P = 0.025 (NRP1), *P = 0.031 (NRP2), ***P = 0.0005, ****P < 0.0001. Statistical analysis was performed using the Holm–Sidak *post hoc* comparison test after one-way ANOVA (*n* = 6).

B    Association of SEMA3C and Plexin B1 (Interactions/cell) as detected by the proximity ligation assay (PLA) in DU145 cells treated with either rhIgG$_1$Fc or SEMA3C:Fc, ****P < 0.0001. Statistical analysis was performed using the Mann–Whitney test.

C    PLA association between SEMA3C and Plexin B1 in DU145 KD cells with either siScrambled (siScr) or si-Plexin B1 siRNAs. Cells were treated with either SEMA3C or control (rhIgG$_1$Fc), ****P < 0.0001. Statistical analysis was performed using the Mann–Whitney test.

D    PLA interactions of SEMA3C and NRP1 in control (rhIgG$_1$Fc) or SEMA3C:Fc-treated DU145 cells transfected with either siPLXNB1 or siScr, ****P < 0.0001. Statistical analysis was performed using the Mann–Whitney test.

E    SEMA3C or rhIgG$_1$Fc as control DU145 lysates were immunoprecipitated with antibodies against NRP-1, PLEXIN B1, HER2/ErbB2, and MET. The immunoblot was probed with C-terminal-specific Plexin B1 antibodies. Input levels of the corresponding proteins are shown.

F–H    PLA association of Plexin B1 and MET (F), EGFR (G), and HER2 (H) in DU145 cells treated with either rhIgG$_1$Fc or SEMA3C:Fc. Negative controls represent the background staining observed in the absence of primary antibody added in the PLA reaction. Representative corresponding fluorescence micrographic images of the PLA interactions of rIgG$_1$Fc versus SEMA3C:Fc-treated DU145 cells are shown. DAPI staining in blue. **P = 0.005, ****P < 0.0001. Data were statistically tested using Kruskal–Wallis or Mann–Whitney test.

Data information: Scale bars: 10 μm. The punctate red fluorescence staining represents the specific corresponding PLA interactions (B–D, F–H). Horizontal lines represent the Mean and the box range represents the minimum and maximum interactions /cell. Data is representative of PLA interactions from 5 fields of view.
Source data are available online for this figure.

    

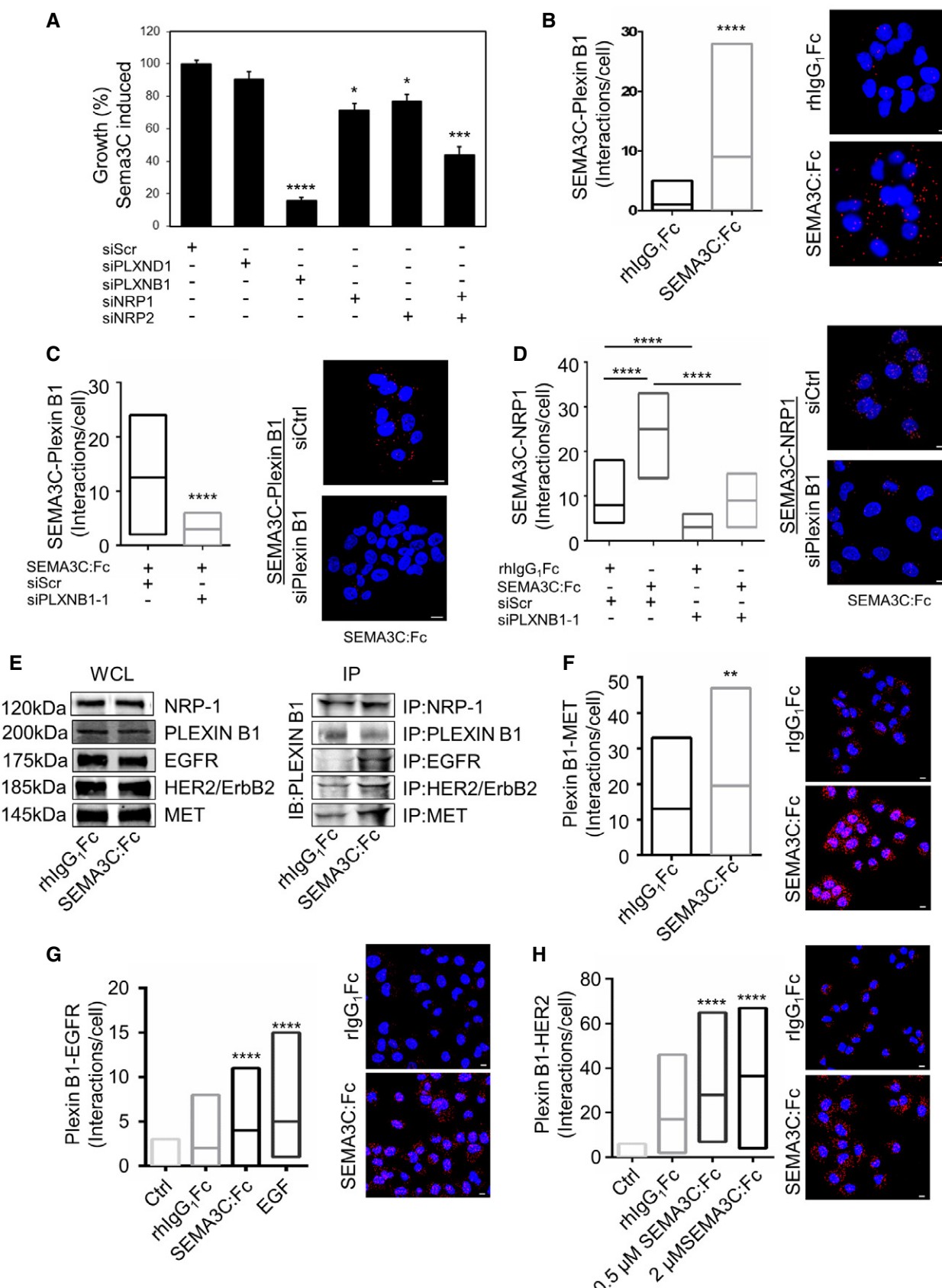

**Figure 2.**

that was significantly inhibited by siPLXNB1 treatment (Fig 2C). Furthermore, we found significant binding of SEMA3C with NRP1 (Fig 2D), whereas siPLXNB1 treatment abrogated the association of SEMA3C with NRP1, suggesting that binding of SEMA3C to NRP1 is Plexin B1-dependent. However, while DU145 cells express low levels of Plexin D1 (Appendix Fig S3A), we observed no interaction between Plexin D1 and SEMA3C in DU145 cells (data not shown).

As a complementary approach, we performed classic ligand binding assays (Flanagan & Leder, 1990) utilizing a SEMA3C-alkaline phosphatase (SEMA3C-AP) fusion protein as an affinity probe on DU145 cells treated with siPLXNB1 versus siScr. Plexin B1 silencing using two distinct siRNA sequences significantly reduced SEMA3C-AP binding to DU145 cells (Appendix Fig S1F). Furthermore, classical co-immunoprecipitation studies also showed an increased association of EGFR, Her2/ErbB2, and MET with Plexin B1 in DU145 cells treated with SEMA3C:Fc compared to controls (Fig 2E). We also observed that SEMA3C stimulation significantly increased complex formation between Plexin B1/MET (Fig 2F), Plexin B1/EGFR (Fig 2G), and Plexin B1/HER2 as detected using PLA in DU145 cells (Fig 2H). We also observed cognate ligand, EGF-mediated complex formation of EGFR and Plexin B1 (Fig 2G).

We next performed Oncomine analysis to explore the potential clinical relevance of Plexin B1 in prostate cancer. Oncomine analysis revealed that transcript levels of Plexin B1 were elevated in prostate cancer samples compared to normal prostate tissue in three independent microarray gene expression profiling studies (Magee et al, 2001; Lapointe et al, 2004; Wallace et al, 2008) (Appendix Fig S3B).

### SEMA3C drives growth and castration resistance

To determine the effects of SEMA3C on growth and castration resistance of PCa cells, LNCaP cells (which express low levels of SEMA3C) were stably transduced with a lentiviral vector (Naldini et al, 1996) expressing full-length SEMA3C (LNCaP$_{SEMA3C-FL}$) or an empty vector (LNCaP$_{empty}$) as a control. SEMA3C-overexpressing LNCaP$_{SEMA3C-FL}$ cells showed enhanced cell growth compared to LNCaP$_{empty}$ cells (Fig 3A). SEMA3C overexpression also led to increased cell growth in androgen-free serum (CSS) conditions following 72 h of culture (Fig 3B) and an augmented growth response to increasing concentrations of synthetic androgen R1881 (Fig 3B). Enhanced growth of LNCaP$_{SEMA3C-FL}$ cells correlated with an increased proportion of cells in S phase and decreased proportion of cells in G1 as compared to LNCaP$_{empty}$ cells consistent with accelerated G1-S transition of the cell cycle of LNCaP$_{SEMA3C-FL}$ cells (Fig 3C). Compared to LNCaP$_{empty}$ cells, LNCaP$_{SEMA3C-FL}$ cells exhibit an increased proportion of cells undergoing DNA synthesis as revealed by incorporation of BrdU-FITC (Fig 3D) and under serum-reduced conditions, a significant decrease of cells with sub-G0/G1 DNA content upon treatment with recombinant SEMA3C:Fc compared to PBS control, suggesting that SEMA3C treatment inhibits apoptosis (Fig 3E). We next evaluated whether SEMA3C overexpression confers castration-resistant growth in vivo. The rates of tumor growth in vivo after castration were compared between LNCaP$_{SEMA3C-FL}$ (n = 10) and LNCaP$_{empty}$ (n = 10) tumors. Under castrate conditions, LNCaP SEMA3C-FL tumors exhibited enhanced tumor growth rates as compared to LNCaP empty controls, implying that SEMA3C promotes castration-resistant growth (Fig 3F).

Next, we assessed the effects of SEMA3C overexpression on tumor incidence and tumor growth of LNCaP xenografts implanted

orthotopically in prostates of SCID mice under eugonadic conditions (Fig 3G; Sato et al, 1997). All mice were sacrificed 12 weeks after inoculation of tumors. Mice were orthotopically inoculated with $1 \times 10^6$ cells which generate a sub-plateau tumor–take rate that allows the ability to better distinguish between control and experimental groups. LNCaP$_{SEMA3C-FL}$ cells showed higher tumor incidence than LNCaP$_{empty}$ group; tumors were observed in 89% (16 out of 18) of mice in the LNCaP$_{SEMA3C-FL}$ group and 29% (four out of 14) of mice in the LNCaP$_{empty}$ group. These data suggest that SEMA3C may be a key driver of prostate cancer growth and development in vivo.

### Autocrine SEMA3C signaling drives prostate cancer cell growth

High SEMA3C-expressing DU145 cells exhibit constitutive activation of EGFR pathway, suggesting RTK pathway activation may be occurring through potential autocrine signaling loop whereby secreted SEMA3C binds to autocrine receptors, Plexin B1 and NRP1/2, on the cell surface leading to transactivation of EGFR. From immunoblot analyses, we found that SEMA3C silencing in DU145 cells with two different siRNAs (siSEMA3C-1 and siSEMA3C-2) effectively silenced SEMA3C protein expression versus scrambled siRNA controls (Fig 4A–C). SEMA3C knockdown was associated with decreased phosphorylation of EGFR, Her2/ErbB2, MET, SHC, AKT, SRC, and p44/42 MAPK (Fig 4A) and increased apoptosis induction as monitored by increased caspase-3 and PARP cleavage (Fig 4B). Moreover, SEMA3C silencing suppressed cell growth that was rescued by the addition of recombinant SEMA3C: Fc (Fig 4C).

Next, to determine whether RTK pathway activation in DU145 cells was Plexin B1-dependent, we treated DU145 cells with siPLXNB1 or siScr and examined RTK pathway activation by immunoblotting. Our results showed that Plexin B1 knockdown inhibited EGFR, MET, and SHC phosphorylation in DU145 cells (Fig 4D). Furthermore, SEMA3C-induced EGFR and SHC phosphorylation in LNCaP cells was diminished by Plexin B1 silencing (Appendix Fig S3C). In addition, treatment of LNCaP cells with EGFR inhibitor, erlotinib, inhibited SEMA3C-induced growth response (Appendix Fig S3D) and EGFR signaling (Appendix Fig S3E). Collectively, these data suggest that autocrine SEMA3C drives cancer cell growth and survival via RTK pathway activation.

### SEMA3C ASO delays LNCaP tumor progression after castration in vivo

In order to perform in vivo proof-of-concept validation studies, we designed antisense oligonucleotides targeting nucleotide positions 1–20 of the SEMA3C coding sequence (SEMA3C ASO) for SEMA3C gene silencing in vivo. SEMA3C ASO inhibited SEMA3C mRNA and protein expression in a dose- and sequence-specific manner in castration-resistant C4-2 (Fig 4E and F) cells as compared to scrambled (Scr) oligonucleotide as control. This inhibition also led to a dose-dependent decrease in cell growth (Fig 4G) and inhibition of phosphorylation of EGFR, SHC, and p44/42 MAPK (Fig 4H). The growth inhibitory effect by SEMA3C ASO was rescued by addition of recombinant SEMA3C (Fig 4I). SEMA3C ASO also enhanced apoptosis as shown by increased sub-G0/G1 DNA content (Fig 4J) and an increase in expression of cleaved PARP and cleaved caspase 3 (Fig 4K). Similarly, SEMA3C ASO also inhibited cell growth and

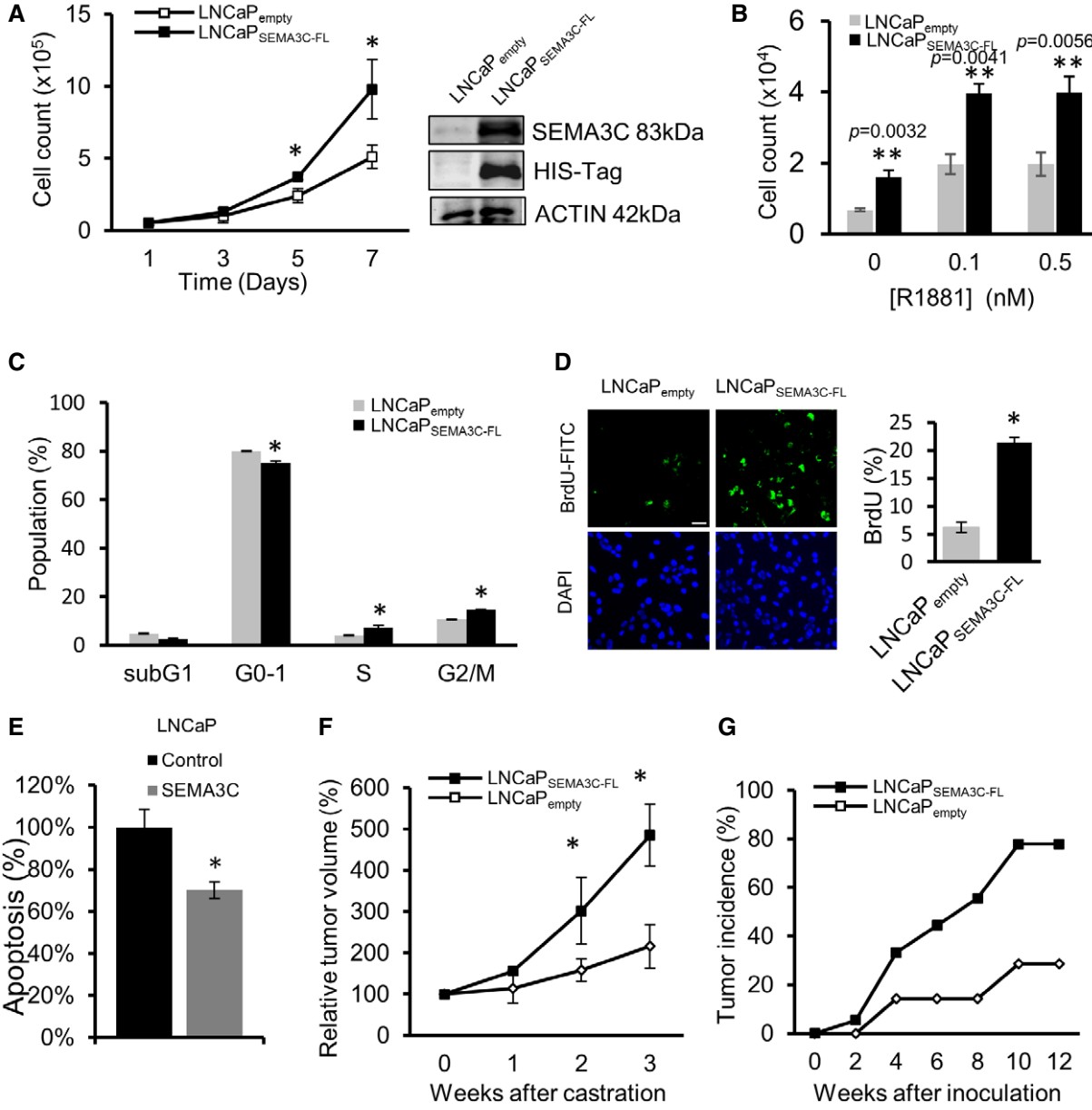

**Figure 3. SEMA3C regulates prostate cancer cell growth.**

A  Growth of LNCaP$_{empty}$ and LNCaP$_{SEMA3C-FL}$ cells cultured in medium containing 5% fetal bovine serum (FBS) as monitored by cell counting, *$P < 0.05$. Inset shows immunoblot of SEMA3C protein levels in LNCaP$_{empty}$ and LNCaP$_{SEMA3C-FL}$ using SEMA3C (N20) and HIS-tag antibodies. Data represents the mean $\pm$ SEM cell count from experiments performed in triplicate.

B  Growth of LNCaP$_{empty}$ and LNCaP$_{SEMA3C-FL}$ cells treated with R1881 (0–0.5 nM) was assessed after 72 h as above. Bars represent the mean $\pm$ SEM cell count from experiments performed in triplicate.

C  Cell cycle analysis of LNCaP$_{empty}$ and LNCaP$_{SEMA3C-FL}$ cells cultured in medium containing 10% FBS. Bars represent the mean proportion of total cells in the indicated cell cycle $\pm$ SEM as determined by propidium iodide staining and flow cytometry, *$P < 0.05$. Bars represent the relative percent mean $\pm$ SEM population from triplicates.

D  Left panels show fluorescence microscopic images of LNCaP$_{empty}$ and LNCaP$_{SEMA3C-FL}$ cells stained with BrdU-FITC and DAPI after 72 h in 0.5% FBS. Scale bar: 50 μm. The right panel shows % BrdU-FITC-positive cells (mean $\pm$ SEM) from five fields of view, *$P < 0.05$.

E  LNCaP cells (2.5 × 10$^5$ cells/well) were treated in RPMI supplemented with 0.5% CSS in the absence (PBS) or presence of SEMA3C:Fc (0.5 μM) for 4 days. The cells were collected for propidium iodide staining and analyzed by flow cytometry for the proportion of cells with sub-Go/G1 DNA content. Bars represent the mean percent of maximum apoptosis relative to the control $\pm$ SEM of treatments performed in triplicate (*$P = 0.016$).

F  Tumor progression *in vivo* after castration was compared between LNCaP$_{SEMA3C-FL}$ and LNCaP$_{empty}$ tumors ($n = 10$). Data represent the relative % tumor volume (mean $\pm$ SEM) up to 3 weeks after castration, *$P < 0.05$.

G  Tumor incidence (%) in SCID mice inoculated with either LNCaP$_{SEMA3C-FL}$ ($n = 18$) or LNCaP$_{empty}$ ($n = 14$) cells up to 12 weeks after inoculation.

Data information: Statistical analysis was carried out using the unpaired two-tailed Student's *t*-test.
Source data are available online for this figure.

**Figure 4.**

**Figure 4.   SEMA3C silencing inhibits CRPC cell growth and induces apoptosis.**

A   DU145 cells transfected with siScr, siSEMA3C-1, or siSEMA3C-2 siRNA. SEMA3C silencing was confirmed by immunoblotting using SEMA3C Abs. The effect of SEMA3C silencing on EGFR signaling is shown by immunoblotting with phospho-specific Abs for EGFR, HER2/ErbB2, MET, SHC, AKT, SRC, and MAPK. The data are representative of independent experiments repeated four times.

B   DU145 cells transfected with siScr, siSEMA3C-1, or siSEMA3C-2 and cultured in medium containing 10% FBS for 72 h. Apoptosis was demonstrated by immunoblotting with cleavage-specific PARP and caspase-3 Abs and with PARP Abs that recognizes both native (116 kDa) and cleaved (89 kDa) PARP. Vinculin is shown as loading control. The data are representative of three independent experiments.

C   24 h after transfection with siScr, siSEMA3C-1, or siSEMA3C-2, DU145 cells were treated ±1 μM SEMA3C:Fc and cell growth was determined at 48 h using PrestoBlue proliferation assay. Data represent the mean ± SEM of triplicate. Inset shows SEMA3C expression level in cells after 48 h in culture.

D   EGFR, MET, and downstream SHC phosphorylation in DU145 cells treated with Plexin B1 siRNA. Plexin B1 knockdown and vinculin levels are shown as controls. Data are representative of four independent experiments.

E   Relative SEMA3C mRNA levels in C4-2 cells transfected with mock (control), scrambled oligonucleotides (Scr), or SEMA3C ASO at the indicated doses (nM). Bars represent the mean relative quantity (RQ) mRNA ± SEM from samples performed in biological triplicate each analyzed in triplicate.

F   SEMA3C ASO induces dose- and sequence-specific silencing of SEMA3C. C4-2 cells were transfected with mock, Scr, or SEMA3C ASO at the indicated doses (nM). SEMA3C expression levels were determined by immunoblot analyses.

G   The growth (%) of C4-2 cells transfected with Scr versus SEMA3C ASO at the indicated doses (nM) was assayed in triplicate at day 3 using the CyQUANT proliferation assay. Data represent the mean cell growth ± SEM expressed as percentage of control on day 0, **$P < 0.005$. Statistical analysis was performed using the unpaired two-tailed Student's *t*-test (assayed in triplicate).

H   Levels of EGFR, SHC, and MAPK phosphorylation were determined by immunoblot analyses of cell lysates from C4-2 cells transfected with either mock, Scr, or SEMA3C ASO at the indicated concentrations (nM). Data are representative of three independent experiments.

I   LNCaP cells transfected with either Scr or SEMA3C ASO (100 nM) were cultured ± SEMA3C:Fc. Growth (%) was assayed by PrestoBlue proliferation assay. Bars represent relative growth FI (fluorescence intensity) after 48 h relative to day 0. Data represent the mean cell growth ± SEM of control ScrASO versus control SEMA3CASO, **$P = 0.0062$; control ScrASO versus SEMA3C-treated ScrASO, *$P = 0.0423$; and control SEMA3CASO versus SEMA3C-treated SEMA3CASO, **$P = 0.004$ (upper panel). Statistical analysis was performed using the unpaired two-tailed Student's *t*-test. Immunoblot showing the rescue of EGFR phosphorylation and SEMA3C ASO knockdown (lower panel).

J   C4-2 cells were treated with either Scr or SEMA3C ASO at the indicated doses. Bars represent the percentage of cells in the sub-G1 population (mean ± SEM) from triplicate samples as determined by PI staining and flow cytometric analysis.

K   Apoptosis was assessed by immunoblotting with Abs to cleaved PARP, cleaved caspase-3, and actin on cell lysates from C4-2 cells 72 h after transfection with either oligofectamine, Scr, or SEMA3C ASO.

Source data are available online for this figure.

induced apoptosis of a second castration-resistant cell line, CWR22Rv1 (Appendix Fig S4A–F).

Using SEMA3C ASO to achieve *in vivo* SEMA3C gene silencing, we tested whether SEMA3C inhibition could delay time to CRPC progression of LNCaP xenografts post-castration (Gleave *et al*, 1992). Twenty male athymic nude mice bearing LNCaP xenograft tumors were castrated and randomly selected for treatment with SEMA3C ASO versus Scr control when serum prostate-specific antigen (PSA) levels reached a threshold of 75 ng/ml. Mean tumor volume and PSA levels were similar in both groups at the beginning of the treatment. Beginning 1 day after castration, 12.5 mg/kg of ASO was administered every other day by intraperitoneal (i.p.) injection for 6 weeks and tumor volume and serum PSA levels were monitored once weekly. As shown in Fig 5A and B, LNCaP tumor volume and serum PSA levels decreased following castration and remained low throughout the time course of the experiment in mice treated with SEMA3C ASO as compared with those treated with scrambled controls that exhibited standard kinetics of gradual CRPC progression. All mice treated with castration plus SEMA3C ASO had a significant inhibition of CRPC tumor growth during the 6 weeks of analyses. No treatment-related adverse side effects (i.e., changes in body weight, gross morphology, and behavior) were observed with SEMA3C ASO or Scr control treatment. In addition, tumors treated with SEMA3C ASO showed decreased intensity of SEMA3C, CD31, and Ki67 staining, with increased intensity of TUNEL staining (Fig 5C).

### SEMA3C ASO treatment suppresses established CRPC tumor growth *in vivo*

We next evaluated the effect of SEMA3C ASO treatment on the growth of established CRPC tumors *in vivo*. Male nude mice bearing

LNCaP CRPC tumors (> 150 mm³) were randomly assigned and treated with 12.5 mg/kg of Scr ASO or SEMA3C ASO by i.p. injection. While mean CRPC tumor volume at baseline was similar in each group, SEMA3C ASO treatment significantly attenuated CRPC tumor growth (Fig 5D) and serum PSA levels (Fig 5E). No apparent adverse effect by SEMA3C ASO was observed, and body weight in each group was not significantly different during the treatment (data not shown). Tumor and PSA doubling times were extended after SEMA3C ASO therapy as compared to Scr controls (Fig 5F). SEMA3C ASO-treated tumors have suppressed levels of SEMA3C, Ki-67, PSA, and AR expression as well as higher apoptotic rates as shown by increased TUNEL staining compared to Scr control-treated tumors (Fig 5G).

### SEMA3C ASO inhibits enzalutamide-resistant prostate cancer cell growth

Current therapies available for treatment of CRPC include enzalutamide (ENZ), a potent AR antagonist against which resistance can arise. We next evaluated whether SEMA3C ASO could inhibit growth of ENZ-resistant prostate cancer. We compared SEMA3C expression between ENZ-resistant and ENZ-sensitive PCa tumors (Kuruma *et al*, 2013; Matsumoto *et al*, 2013). As shown in Fig 6A, from initial analyses of a limited number of ENZ-resistant tumors, increased expression of SEMA3C, AR, and PSA was observed in some ENZ-resistant tumors. SEMA3C ASO treatment decreased SEMA3C, AR, and PSA protein expression (Fig 6B), induced PARP cleavage (Fig 6C), and decreased cell growth (Fig 6D). Moreover, SEMA3C ASO suppressed ENZ-resistant MR49F xenograft tumor growth *in vivo* as monitored by tumor volume and serum PSA levels (Fig 6E and F).

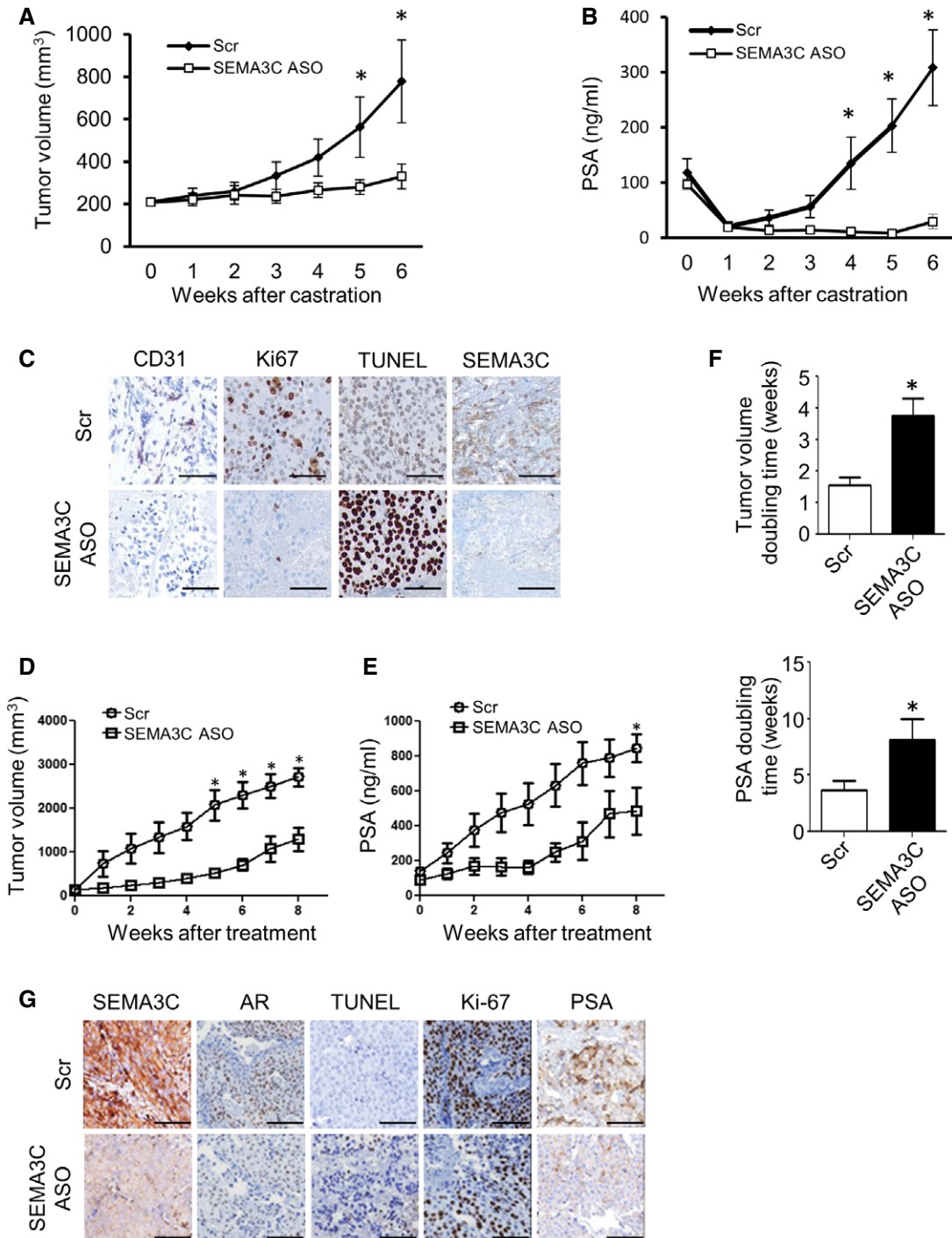

**Figure 5. SEMA3C ASO treatment delays tumor progression after castration and suppresses CRPC tumor growth *in vivo*.**

A, B Tumor volumes (mm³) (A) and serum PSA levels (ng/ml) (B) after castration of LNCaP xenograft tumor-bearing athymic *nu⁻/⁻* mice treated with either Scr or SEMA3C ASO (12.5 mg/kg). 10 male mice were treated in each group.

C Photo-micrographs of tumor tissue sections stained with CD31, Ki-67, TUNEL, and SEMA3C, treated with either Scr or SEMA3C ASO.

D, E Tumor progression in male nude mice harboring LNCaP CRPC tumors (> 150 mm³) that were treated with either Scr or SEMA3C ASO by intraperitoneal injection. CRPC tumor volume (mm³) (D) and PSA levels (ng/ml) (E) over 8 weeks of treatment. 10 male mice were treated in each group.

F Tumor volume (mm³) and PSA doubling times (weeks) of tumors from nude mice harboring LNCaP CRPC xenografts that were treated with either Scr or SEMA3C ASO by i.p. injection.

G Photo-micrographs of CRPC tumor tissue sections stained with SEMA3C, AR, TUNEL, Ki-67, and PSA.

Data information: Scale bars: 100 μm. Data represent mean ± SEM; *$P < 0.05$. Statistical analysis was performed using the Mann–Whitney test. 10 male mice were treated in each group.

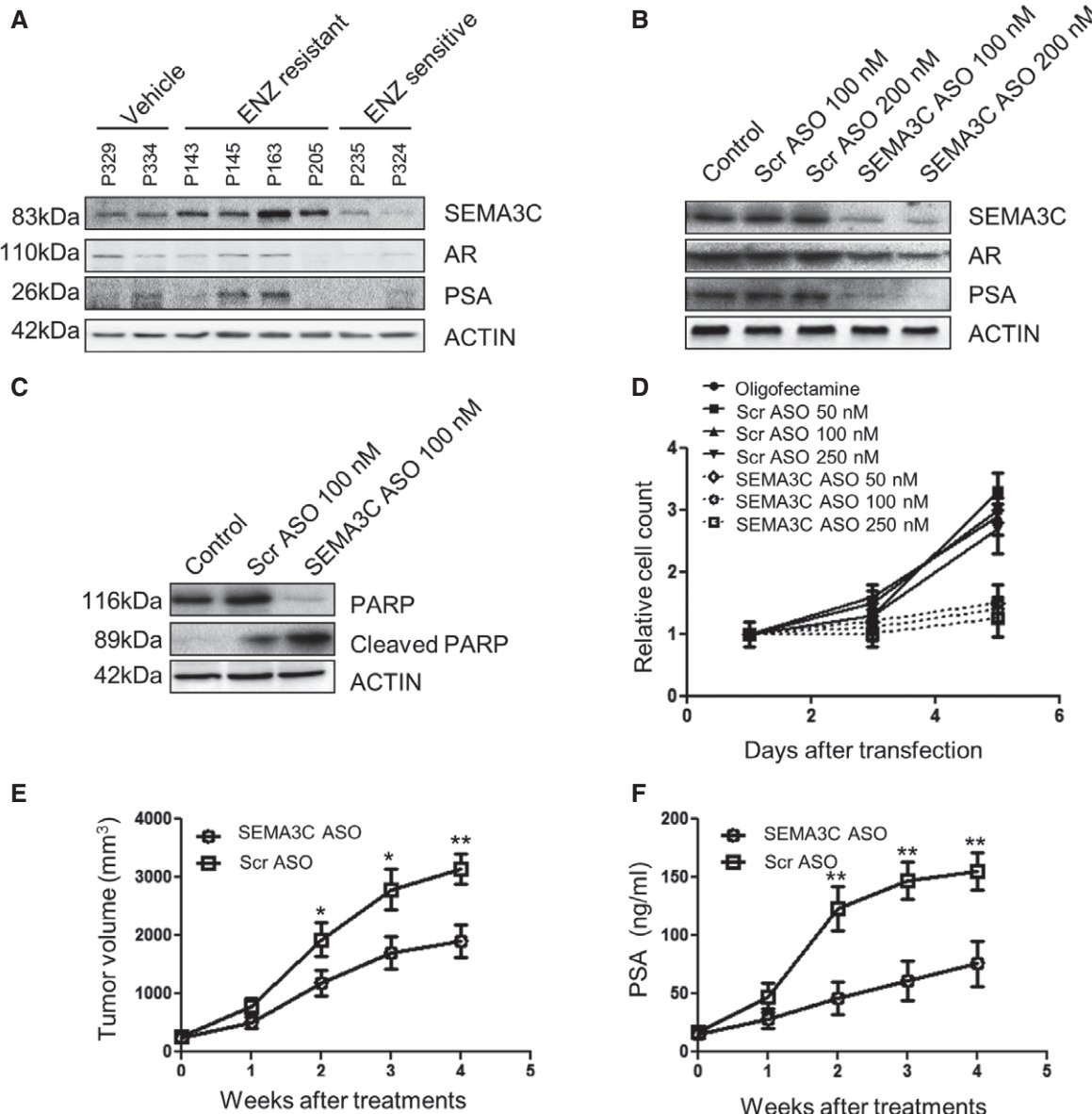

**Figure 6.  SEMA3C inhibition suppresses growth of ENZ-resistant PCa.**

A   Immunoblot analyses showing levels of SEMA3C, AR, and PSA from ENZ-resistant and ENZ-sensitive tumor tissue obtained from mice harboring LNCaP xenograft tumors treated with either vehicle or ENZ (10 mg/kg), $n$ = 15 per group.

B   MR49F cells transfected with either mock, Scr or SEMA3C ASO at the indicated concentrations and cultured for 48 h. SEMA3C, AR, and PSA levels were examined from total protein lysates by immunoblot analyses.

C   MR49F cells were transfected with either mock, Scr, or SEMA3C ASO (100 nM). Cells were lysed after 48 h, and PARP cleavage was examined by immunoblotting.

D   Cell growth of MR49F cells transfected with either mock or the indicated concentration of Scr or SEMA3C ASO (nM). Relative cell numbers were determined by direct cell count from triplicate samples.

E, F   Tumor growth (mm³) (E) and PSA levels (ng/ml) (F) were monitored in mice bearing MR49F tumors treated with Scr or SEMA3C ASO (12.5 mg/kg) over a period of 4 weeks. Each experimental group consisted of 7 mice.

Data information: Data represent mean ± SEM; *$P$ < 0.05, **$P$ < 0.001. Statistical analysis was performed using the Mann–Whitney test.
Source data are available online for this figure.

## Development of a therapeutic protein inhibitor of Semaphorin 3C–Plexin B1 signaling

As shown above, we have found that SEMA3C-induced RTK pathway activation was dependent on Plexin B1. With the aim of developing a therapeutic protein inhibitor of SEMA3C signaling, we engineered a Plexin B1:Fc fusion decoy protein to functionally disrupt SEMA3C-induced RTK activation. We generated two independent lentiviral constructs, depicted in Fig 7A the first containing the Plexin B1 SEMA domain and adjacent PSI fused to a linker

sequence and Fc domain of human IgG1 (B1SP). The second construct contained an extracellular fragment of Plexin B1 comprising N-terminal sema domain, 3 PSI and 3 IPT domains up to the first proprotease convertase cleavage site at position 1,302 of the Plexin B1 similarly fused to a linker and human IgG1 Fc (B1R4, Fig 7A) which is a naturally occurring cleaved extracellular fragment of Plexin B1 that is generated upon proteolytic processing by proprotein convertases (Artigiani *et al*, 2003). Although both fusion proteins were expressed efficiently in lentiviral-transduced cells, only B1SP recombinant protein was effectively secreted (Fig 7B and C). We therefore focused on B1SP as a viable inhibitor of SEMA3C: Plexin B1 signaling. Next, we sought to validate whether B1SP could interact with NRP1 and Plexin B1. Co-immunoprecipitation experiments from lysates of DU145 cells treated with B1SP (0–2 μM) or recombinant human IgG:Fc as control showed that B1SP binds to NRP1 and Plexin B1 in a dose-dependent manner (Fig 7D) using C-terminal-specific Plexin B1 antibodies to detect endogenous Plexin B1. We next conjugated phycoerythrin to B1SP and used this reagent to perform cell binding assays using flow cytometry of DU145 cells treated with varying doses of B1SP:PE. Our results showed that B1SP bound to DU145 cells in a dose-dependent manner with $EC_{50}$ = 8.5 nM (Appendix Fig S5A). In a complementary approach, we used PLA to demonstrate the association between B1SP and Plexin B1 (Fig 7E). Next we sought to determine whether B1SP could inhibit the binding of SEMA3C with its receptors NRP1 and Plexin B1. Our PLA results showed that B1SP significantly inhibited the interaction between SEMA3C and Plexin B1 (Fig 7F) and between SEMA3C and NRP1 (Fig 7G).

### B1SP inhibits cell proliferation of PCa cells, SEMA3C-induced RTK activation, and delays *in vivo* tumor progression of LNCaP xenografts post-castration

To determine whether B1SP could be a potential therapeutic for PCa treatment, we performed proliferation assays in androgen-sensitive LNCaP, and castration-resistant DU145, C4-2, and 22Rv1. We first demonstrated that B1SP could inhibit SEMA3C-induced proliferation of LNCaP cells (Fig 8A). In order to address the specificity of B1SP to inhibit cell growth, we designed an analogous fusion protein containing the Plexin D1 SEMA domain and adjacent PSI fused to a linker sequence and Fc domain of human IgG1 (D1SP) (Appendix Fig S5B). As compared to B1SP, D1SP was an ineffective inhibitor of LNCaP cell growth (Appendix Fig S5C). B1SP also inhibited cell growth of C4-2, DU145, and 22Rv1 cells (Fig 8B). Moreover, B1SP inhibited R1881-induced cell growth of LNCaP in a dose-dependent manner (Fig 8C), suggesting that B1SP could inhibit androgen-induced PCa growth consistent with our recent report that SEMA3C is an androgen-induced gene (Tam *et al*, 2016). Next we performed similar cell growth assays on a panel of cell lines representing kidney cancer, bladder cancer, and glioblastoma. The $IC_{50}$ value for the panel of cell lines is shown in Appendix Table S2. Our results show that in addition to PCa, B1SP inhibits cell proliferation of various tumor cell types (Appendix Fig S5D). B1SP inhibited SEMA3C-induced activation of EGFR, Her2/ErbB2, and MET pathways in DU145 cells (Fig 8D), and SEMA3C-induced activation of EGFR pathway in LNCaP (Fig 8E) and C4-2 cells (Fig 8F). We did not observe any inhibitory activity of SEMA3C-induced activation of RTK signaling by D1SP treatment of LNCaP (Appendix Fig S5E) or

DU145 cells (Appendix Fig S5F). B1SP inhibited EGF-induced EGFR pathway activation in LNCaP (Fig 8G), C4-2 (Fig 8H), CWR22Rv1 (Appendix Fig S5G), and MR49F (Appendix Fig S5H). B1SP also inhibited SEMA3C- induced phosphorylation of MET pathway activation in Caki-2 and T24 cells (Appendix Fig S5I and J) and HGF-induced phosphorylation of MET and MAPK in T24 cells (Appendix Fig S5K). Taken together, these results demonstrate that B1SP inhibits SEMA3C-induced activation of multiple RTK signaling as well as EGF- and HGF-induced activation of EGFR and MET, respectively.

We next analyzed whether B1SP could be utilized therapeutically to delay progression of LNCaP tumor xenografts post-castration *in vivo*. To this end, athymic $nu^{+}$ mice bearing LNCaP xenografts were castrated and injected intraperitoneally (ip) with purified B1SP protein or PBS as control. We first examined biodistribution and pharmacodynamics (PD) of B1SP. To test whether B1SP was present in the tumor tissue of LNCaP xenograft tumors, protein lysates and tumor sections were prepared from the tumor tissue harvested from control (PBS)- or B1SP-treated mice. We observed anti-HIS-tag-specific staining in B1SP-treated mice compared to control in IHC sections. Moreover, we observed a trend in the reduction of MAPK phosphorylation as a PD endpoint in B1SP-treated xenograft IHC sections compared to vehicle-treated tumor tissue (Appendix Fig S6A). In addition, we observed reduced phosphorylation of MAPK in B1SP protein by Western blot analysis (Appendix Fig S6B). Histologic examination of the tumors also revealed reduced cellularity and proliferation by H&E and Ki-67 staining, respectively, in B1SP-treated compared with vehicle-treated tumors (Appendix Fig S6C). Treatment of LNCaP tumor-bearing mice with B1SP significantly delayed castrate-resistant re-growth post-castration as monitored by tumor volume (Fig 8I) and serum PSA (Fig 8J). Furthermore, B1SP treatment inhibited tumor growth of established castrate-resistant C4-2 xenografts as monitored by tumor volume (Appendix Fig S6D) and serum PSA levels (Appendix Fig S6E). Treatment with B1SP had no effect on body weight as compared to PBS-treated control mice.

## Discussion

In this study, we found that SEMA3C is a secreted soluble autocrine growth factor that promotes growth of PCa cancer via Plexin B1- and NRP1-mediated transactivation of EGFR, HER2/ErbB2, and MET signaling with activation of downstream signaling pathways such as SRC, MAPK, and PI3K/AKT pathways. We report that high SEMA3C expression is associated with metastatic CRPC and with RTK pathway activation. Moreover, SEMA3C overexpression promotes orthotopic tumor growth and confers castrate-resistant growth of LNCaP tumors. Inhibition of SEMA3C inhibits both CRPC- and ENZ-resistant progression, suggesting that SEMA3C antagonists may represent a new therapeutic strategy for treatment of CRPC. These data collectively implicate autocrine SEMA3C signaling as one mechanism for castration and treatment resistance in PCa, and mark SEMA3C signaling as a therapeutic target.

Development of CRPC is a complex process attributed to a variety of mechanisms including activation of androgen receptor and growth factor signaling pathways (Yap *et al*, 2011; Wyatt & Gleave, 2015). We have recently shown that SEMA3C is an androgen

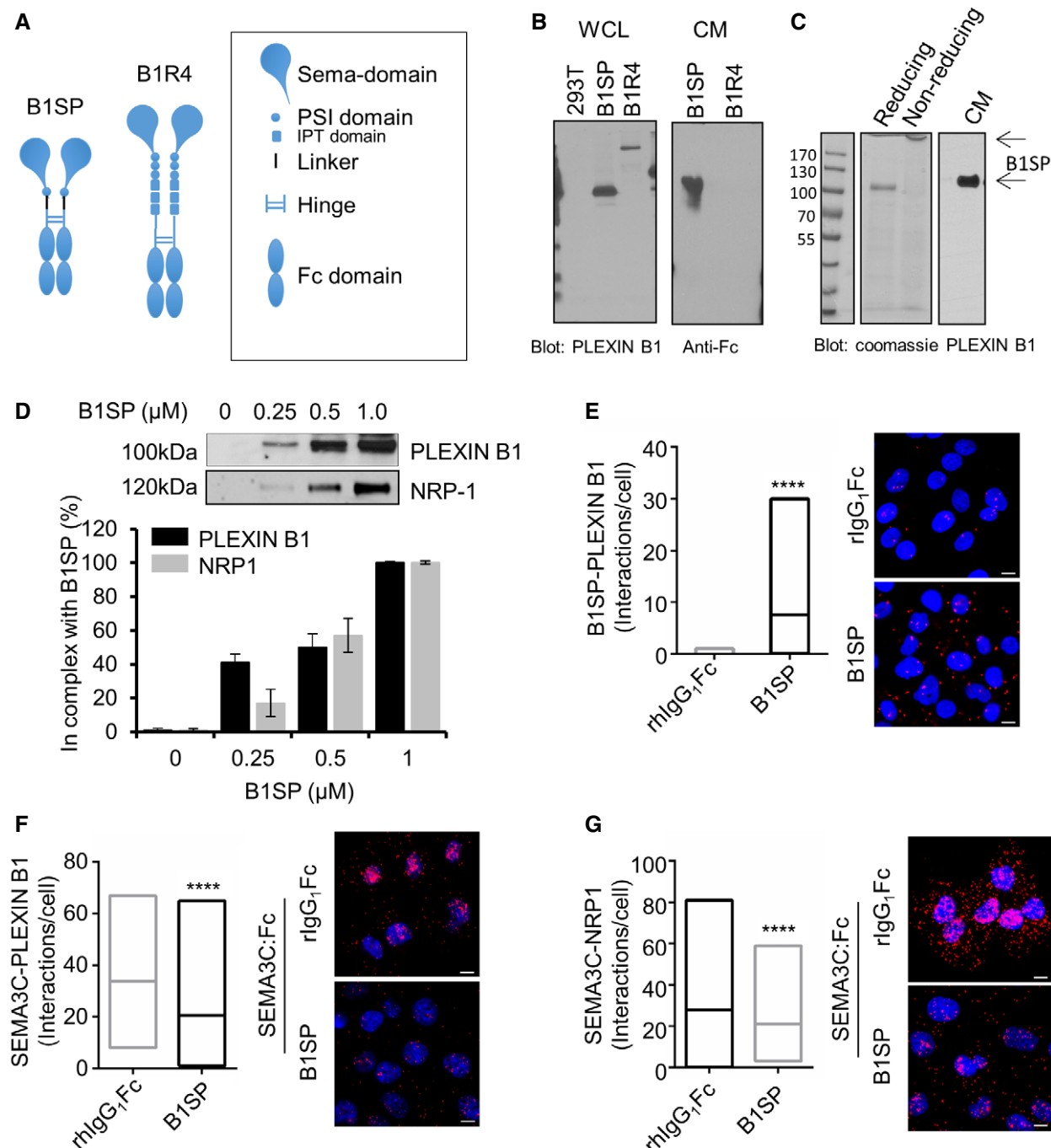

**Figure 7.  Recombinant Plexin B1 inhibitor protein B1SP associates with NRP1 and Plexin B1.**

A   Graphic depicting B1SP and B1R4 recombinant Plexin B1 proteins.

B   Expression of B1SP and B1R4. Western blots show expression of B1SP and B1R4 using Plexin B1 and hIgGFc-specific antibodies.

C   Coomassie staining of B1SP under reducing and non-reducing conditions (left panel) and B1SP secreted CHOS-conditioned medium detected using Plexin B1 antibodies.

D   Dose-dependent binding of B1SP with NRP1 and with endogenous Plexin B1 from DU145 cell lysates. Bars represent the mean ± SEM (%) B1SP in complex with endogenous Plexin B1 or NRP1 from triplicate samples.

E   PLA showing the association between B1SP and Plexin B1 in DU145 cells, ****$P < 0.0001$.

F   PLA association between SEMA3C and Plexin B1 in B1SP- or rhIgGFc-treated DU145 cells that were stimulated with SEMA3C, ****$P < 0.0001$.

G   PLA association between SEMA3C and NRP1 in DU145 cells treated with B1SP or rhIgGFc that were stimulated with SEMA3C, ****$P < 0.0001$.

Data information: Scale bar: 10 μm. Horizontal lines within the boxes represent the mean and the box range represents the minimum and maximum interactions/cell. Data is representative of PLA interactions from 5 fields of view. Statistical analysis was performed using the Mann–Whitney test.
Source data are available online for this figure.

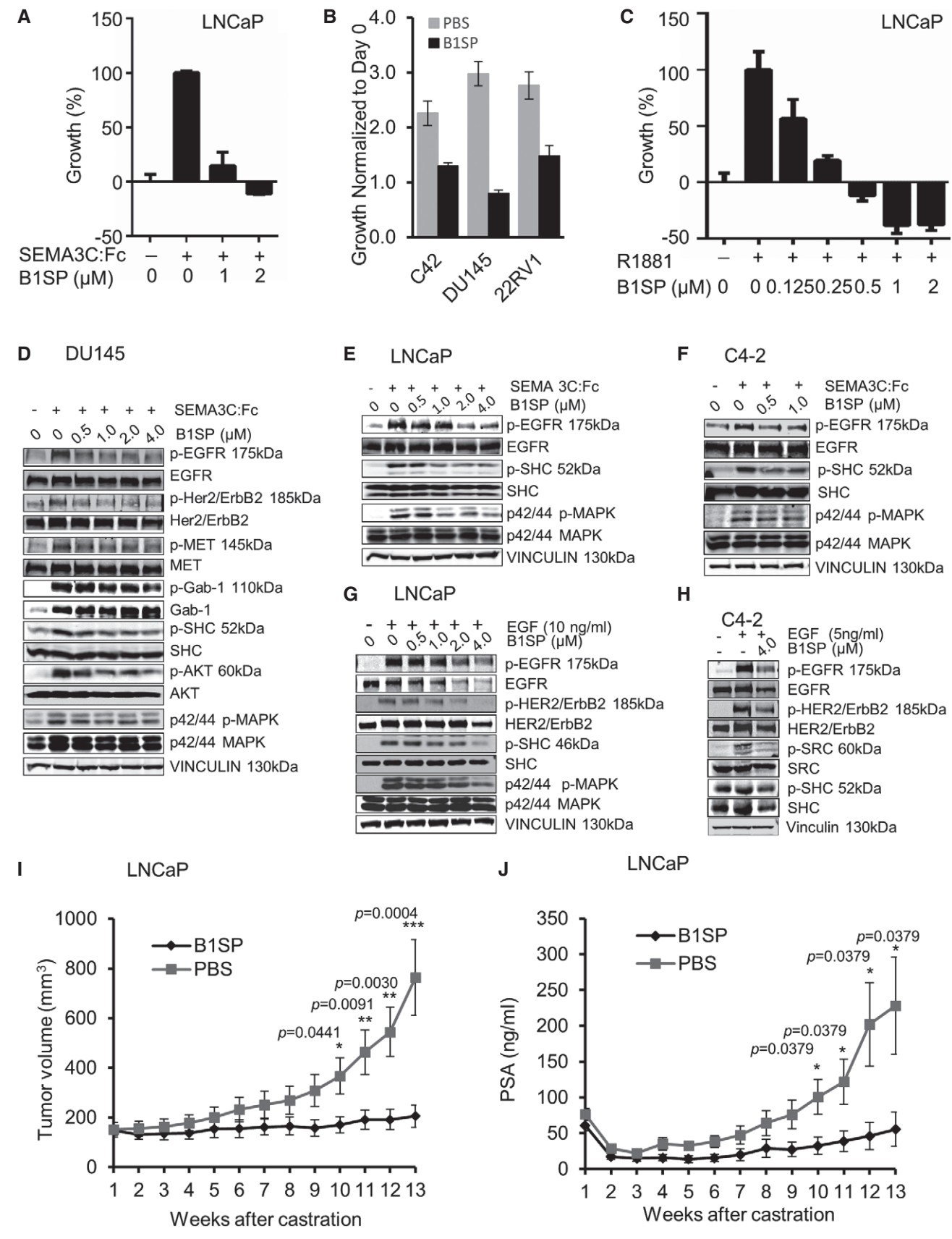

Figure 8.

**Figure 8.  B1SP inhibits cell growth, RTK signaling, and *in vivo* tumor growth.**

A    SEMA3C-induced growth of LNCaP cells treated with B1SP (0–2 μM). Bars represent the mean ± SEM percent growth of LNCaP cells treated in triplicate.
B    Relative growth of C4-2, DU145, and 22RV1 cells treated with 2 μM B1SP or PBS as control for 4 days. Bars represent the mean fluorescence intensity (FI) ± SEM as measured from triplicate wells.
C    R1881 (1 nM)-induced growth of LNCaP cells treated with B1SP (0–2 μM) or ethanol control. Bars represent the % of maximal R1881-induced growth treated in triplicate on day 4.
D    Phosphorylation and total levels of EGFR, Her2/ErbB2, MET, Gab-1, SHC, AKT, and MAPK of +/− SEMA3C-stimulated, B1SP-treated DU145 cells. Equal loading is shown using vinculin antibodies.
E    Phosphorylation and total levels of EGFR, SHC, and MAPK of +/− SEMA3C-stimulated, B1SP (0.5–4 μM)-treated LNCaP cells. Loading controls are shown using vinculin antibodies.
F    Phosphorylation and total levels of EGFR, SHC, and MAPK of +/− SEMA3C-stimulated, B1SP (0–1 μM)-treated C4-2 cells. Loading controls are shown using vinculin antibodies.
G    Phosphorylation and total levels of EGFR, Her2/ErbB2, SHC, and MAPK of +/− EGF-stimulated, B1SP (0–4 μM)-treated LNCaP cells. Vinculin levels are shown as loading controls.
H    Phosphorylation and total levels of EGFR, Her2/ErbB2, SRC, and SHC of +/− EGF-stimulated, C4-2 cells treated with PBS (−) or B1SP (4 μM). Loading controls are shown using vinculin antibodies.
I, J    Tumor volume (mm$^3$) (I) and PSA (ng/ml) (J) from athymic $nu^{-/-}$ mice bearing LNCaP tumors treated with either PBS or B1SP post-castration over a period of 13 weeks. Data are mean ± SEM, $n = 7$. The tumor volume and PSA levels of the treatment groups reached statistical significance 10 weeks after castration. Statistical analysis was performed using the Mann–Whitney test.

Source data are available online for this figure.

receptor-regulated gene that is induced by androgens in prostate cancer (Tam *et al*, 2016). Here, we show that SEMA3C drives activation of multiple RTK pathways and promotes growth and castration resistance. High SEMA3C expression in AR-driven CRPC may be achieved through mechanisms involving reactivation of the AR pathway such as AR gene amplification, AR variant expression, intratumoral steroidogenesis, and AR mutations leading to AR promiscuity. Alternatively, constitutively high SEMA3C expression in non-AR-driven (or AR indifferent) CRPC may be achieved through a bypass mechanism wherein SEMA3C expression is no longer dependent on AR pathway for expression such as in AR-negative DU145 cells.

As shown in Fig 5C, we have found that SEMA3C ASO treatment resulted in reduced levels of the endothelial marker CD31 in LNCaP xenografts post-castration, suggesting that reduced tumor growth may in part be mediated by anti-angiogenic effects of SEMA3C inhibition. SEMA3C has been implicated in angiogenesis and lymphangiogenesis with differing effects depending on cellular context (Serini & Tamagnone, 2015). Genetic studies clearly indicate that SEMA3C plays a distinct role in endothelial cell guidance and vascular morphogenesis (Epstein *et al*, 2015). SEMA3C-deficient mice exhibit severe outflow tract abnormalities such as persistent truncus arteriosus, aortic arch interruption, and mispatterning of intersomitic vessels (Feiner *et al*, 2001). Moreover, SEMA3C has been shown to regulate vascular endothelial cells in a paracrine fashion to control cardiac outflow tract septation (Plein *et al*, 2015). Furthermore, Banu *et al* (2006) have found that SEMA3C promotes endothelial cell survival, adhesion, proliferation, migration, and tube formation *in vitro* that is remarkably similar to the well-established pro-angiogenic factor VEGF-A. However, two recent studies suggest that SEMA3C may exhibit inhibitory activity on pathologic blood vessel formation (Yang *et al*, 2015) and in lymphatic endothelial cells and vascular network in tumor progression (Mumblat *et al*, 2015). As postulated by Serini and Tamagnone, ECs from different vascular beds might display divergent biological responses to SEMA3C, possibly due to the differing cell surface receptor expression profiles (Serini & Tamagnone, 2015). Alternatively, the apparent differential activity of SEMA3C may possibly be due to different expression profile of proteases in the

microenvironment of different vascular beds and subsequent proteolytic processing of SEMA3C. Yang *et al* (2015) found that Sema3CΔ13 isoform, which may result from metalloproteinase cleavage, showed potent anti-angiogenic activity. Similarly, the furin-resistant SEMA3C mutant (FR-Sema3C) construct used by Mumblat *et al* (2015) and Toledano *et al* (2016) contains a C-terminal deletion corresponding to the Sema3CΔ13 isoform that may account for the anti-angiogenic effects observed in these studies.

Here, we show that Plexin B1 is a SEMA3C receptor in PCa cells. These data are consistent with findings by Rohm *et al* (2000) that showed that Plexin A1, A2, and B1 in association with NRP1/2 are receptors for SEMA3C through classical transfection and binding studies. Mouse genetics and other studies have found that Plexins A1, A2, and D1 in association with Nrp1 and Nrp2 are receptors for SEMA3C in endothelial cell and glioma stem cells (Epstein *et al*, 2015; Serini & Tamagnone, 2015). Interestingly, Plexin B1 has been reported by Wong *et al* to be frequently mutated and overexpressed in PCa (Wong *et al*, 2007). Consistent with this, from Oncomine analyses of multiple data sets, Plexin B1 transcript levels were elevated in prostate cancer versus normal prostate tissue samples. Notably, the Plexin B subfamily forms a multireceptor complex via ectodomain interactions with RTKs such as ErbB2 (HER2/neu), and HGF/scatter factor receptor (MET), both of which have been implicated in PCa progression and metastasis (Solit & Rosen, 2007; Cagnoni & Tamagnone, 2014; Wozney & Antonarakis, 2014). Importantly, binding of semaphorins to plexins can transactivate the tyrosine kinase activity of these RTKs independently of their cognate ligands (Giordano *et al*, 2002; Swiercz *et al*, 2004, 2008). Here, we report that Plexin B1 also associates with EGFR and SEMA3C can stimulate activation of EGFR pathway as well as HER2/ErbB2 and MET pathways via binding to Plexin B1/NRP1. Furthermore, we show that SEMA3C drives activation of distinct combinations of RTK signaling pathways depending on the RTK expression profile of the cancer cells and on the context of cell type. For example, SEMA3C drives activation of EGFR in MET-negative LNCaP cells whereas SEMA3C activates EGFR, Her2/ErbB2, and MET in DU145 cells.

Our findings have important implications for clinical management of PCa with RTK and tyrosine kinase inhibitors (TKIs) and provide a potential rational explanation for the disappointing

single-agent activity of EGFR inhibitors (erlotinib, gefitinib), SRC inhibitors (dasatinib, KX2-391, saracatinib) as well as anti-HER2-targeted antibody therapeutics (pertuzumab, trastuzumab) in PCa clinical trials despite strong preclinical evidence for a role of these targets in PCa (Solit & Rosen, 2007; Gallick *et al*, 2012). Considering that SEMA3C drives activation of multiple tyrosine kinase pathways such as EGFR, HER2/ErbB2, MET, and SRC, inhibiting one pathway alone may not be sufficient since other compensatory RTK pathways are concurrently activated. Our data suggest that targeted inhibition of SEMA3C or targeting multiple RTK pathways simultaneously may be necessary to achieve meaningful clinical responses in PCa.

We have found that SEMA3C stimulates EGFR and MET signaling in a broad range of cancer cell lines including bladder, and renal cancer as well as glioblastoma, suggesting that SEMA3C may be an important growth factor for a broad spectrum of cancers in addition to PCa. Members of the ErbB family are important targets in cancer therapy. Therapeutic agents targeting EGFR and ErbB2 are now currently used in treating a variety of cancers. However, their effectiveness is limited in some patients because of the development of intrinsic or acquired resistance. Crosstalk between EGFR and other signal transduction pathways represent potential resistance mechanisms to anti-EGFR agents. For example, in NSCLC, MET amplification is found in up to 20% of patients with acquired resistance to EGFR tyrosine kinase inhibitors (Bean *et al*, 2007). Potentiation of EGFR action by MET in lung cancer has been found to be independent of HGF leading to speculation in favor of an intracellular lateral signaling cascade (Dulak *et al*, 2011). However, since SEMA3C is capable of triggering activation of EGFR, ErbB2, and MET RTK pathways, it represents an alternative extracellular crosstalk mechanism leading to activation of these pathways. Activation of alternate SEMA3C-driven RTKs may confer escape mechanisms for acquired resistance to RTK inhibitors such as ErbB2 amplification in breast cancer and MET amplification in colon and lung cancer.

We generated a recombinant Plexin B1 sema domain-Fc fusion protein called B1SP that effectively inhibited SEMA3C-induced RTK signaling and tumor progression *in vivo*. There are two possibilities underlying the mechanism of action of our Plexin B1 inhibitor. Since B1SP is a soluble Plexin B1 protein, it may sequester SEMA3C from binding the NRP1–Plexin B1 receptor complex. Alternatively, B1SP may bind to endogenous Plexin B1 and NRP1 that effectively attenuate Plexin B1 signaling and complex formation with RTKs. Our data favor the latter possibility as B1SP was capable of complex formation with Plexin B1 and NRP1 and effectively inhibited SEMA3C/NRP1 and SEMA3C/Plexin B1 interactions. Given that the SEMA3C interaction with Plexin B1 was NRP1-dependent, a mechanism involving SEMA3C sequestration by B1SP would be unlikely.

Surprisingly, we have found that B1SP is not only able to inhibit SEMA3C-induced RTK activation but B1SP is also able to suppress RTK activation by its cognate ligands, that is, EGF-induced EGFR activation and HGF-induced MET activation. These findings have important implications for the utility of B1SP as a cancer therapeutic agent exhibiting broad multi-RTK inhibiting activity targeting MET, HER2, and EGFR activated by SEMA3C and its cognate ligands. From PLA studies, we found that Plexin B1 is basally complexed with EGFR, HER2, and MET. The ability of B1SP treatment to inhibit cognate ligand-induced RTK activation of MET and EGFR suggests a model wherein B1SP binding to Plexin B1/NRP1 on the cell surface locks Plexin B1-associated RTKs such as EGFR

and MET in a dimerization-incompetent state. In an unactivated state, the EGFR is found in an autoinhibitory configuration where the dimerization interface of EGFR is intermolecularly occluded (Ferguson *et al*, 2003). Binding of monomeric EGF to the EGFR induces a conformational change that exposes the dimerization interface. We suggest a model wherein B1SP binding may lock the EGFR in a dimerization-incompetent configuration preventing EGF-induced RTK activation.

Our studies provide preclinical proof of concept for use of SEMA3C inhibitors for treatment of advanced PCa, either as single agent or in combination with androgen pathway inhibitors or chemotherapy. Data from studies of SEMA3C knockout mice suggest SEMA3C is dispensable in adult mice; while some homozygous SEMA3C mutant mice die shortly after birth from defects in cardiac development, surviving SEMA3C knockout mice are viable, fertile, and grossly indistinguishable from wild-type or heterozygous littermates (Feiner *et al*, 2001). Hence, while SEMA3C is required for organogenesis during embryonic development, SEMA3C deficiency is well tolerated postnatally suggesting SEMA3C is not required in homeostasis of adult tissues and that SEMA3C inhibition may be well tolerated in adults. Furthermore, Plexin B1 knockout mice are viable and fertile, suggesting that inhibiting Plexin B1 signaling may be relatively safe.

In summary, this study identifies SEMA3C as a key secreted, autocrine growth factor that drives PCa growth and castration- and treatment-resistant prostate cancer progression through activation of EGFR, HER2/ErbB2, MET, and c-SRC tyrosine kinase pathways. Inhibition of SEMA3C-induced and cognate ligand-induced RTK pathway activation with B1SP decoy receptor therapeutic protein represents a new therapeutic strategy that can be used as a single agent or in biologically rational combinations with AR pathway inhibitors and/or TKIs for treatment of advanced prostate cancer.

## Materials and Methods

### Study design

The objective of this study was to evaluate SEMA3C as a therapeutic target for advanced prostate cancer and the physiological consequences of loss and gain of function of SEMA3C. The expression of SEMA3C in a TMA representing a total of 280 prostate cancer specimens obtained from Vancouver Prostate Centre Tissue Bank was evaluated to determine association of SEMA3C expression with castration and treatment resistance.

For all human studies, informed consent was obtained from all subjects and experiments conformed to the principals set out in the WMA Declaration of Helsinki and the Department of Health and Human Services Belmont Report.

### Prostate tissue specimens and TMA immunostaining

The H&E slides were reviewed and the desired areas were marked on them and their correspondent paraffin blocks. Three TMAs were manually constructed (Beecher Instruments, MD, USA) by punching triplicate cores of 0.6 mm for each sample. Most tissues were radical prostatectomy specimens collected from 12 BPH, 114 untreated cases, 87 cases with NHT treatment, and 53 cases treated with NHT

and docetaxel, whereas 30 distant metastatic bone tumors were procured from a rapid autopsy program and kindly provided by Dr. Robert L. Vessella, Department of Urology, University of Washington Medical Center, Puget Sound Veterans Health Care Administration, Seattle, WA, USA. Immunohistochemical staining was conducted by Ventana autostainer model Discover XT™ (Ventana Medical System, Tuscan, AZ, USA) with enzyme-labeled biotin–streptavidin system and solvent-resistant DAB Map kit using 1/100 concentration goat polyclonal antibody against SEMA3C. Digital Imaging and scoring method: All stained slides were digitalized with the SL801 autoloader and Leica SCN400 scanning system (Leica Microsystems; Concord, ON, Canada) at magnification equivalent to ×40. The images were subsequently stored in the SlidePath digital imaging hub (DIH; Leica Microsystems) of the Vancouver Prostate Centre. Representative cores were manually identified by (L.F) Pathologist. Values on a four-point scale were assigned to each immunostain. Descriptively, 0 represents no staining by any tumor cells, 1 represents a faint or focal, questionably present stain, 2 represents a stain of convincing intensity in a minority of cells, and 3 represents a stain of convincing intensity in a majority of cells.

### Assessment of *in vivo* tumor growth for LNCaP xenografts

For *in vivo* SEMA3C overexpression in subcutaneous tumor model studies, ~2 × 10⁶ LNCaP, LNCaP$_{SEMA3C-FL}$, or LNCaP$_{empty}$ cells were inoculated s.c. with 0.1 ml Matrigel (Becton Dickinson Labware, Franklin Lakes, NJ, USA) in the flank region of 6- to 8-week-old male athymic nude mice (Harlan Sprague Dawley, Inc., Indianapolis, IN, USA) via a 27-gauge needle under methoxyflurane anesthesia. When mice bearing LNCaP tumors reached a tumor volume of 200 mm³, they were castrated and tumor volume was monitored by caliper measurement and calculated by the formula length × width × depth × 0.5236 (Janik *et al*, 1975). For *in vivo* orthotopic model, ~1 × 10⁶ LNCaP cells were inoculated into the prostate of SCID mice with 0.1 ml Matrigel. LNCaP tumors were considered as castration-resistant after the levels of PSA values and tumor sizes exceeded levels before castration. For LNCaP castration-resistant (CRPC) xenograft model (Gleave *et al*, 1992), LNCaP cells (1–2 × 10⁶) were inoculated subcutaneously with 0.1 ml Matrigel (Becton Dickinson Labware, Franklin Lakes, NJ, USA) in bilateral flank regions of 6- to 8-week-old male athymic nude mice (Harlan Sprague Dawley, Inc., Indianapolis, IN, USA) via a 27-gauge needle under methoxyflurane anesthesia. Body weight, tumor volume, and serum PSA levels were measured once a week. Blood samples were obtained from tail vein incisions, and serum PSA was determined by PSA ELISA test kit (ClinPro International Co. LLC), or, alternatively measured by the Roche Diagnostics Cobas411 immunoassay system which is an automated, random access multichannel analyzer for immunological analysis using enhanced chemiluminescent technology (ECL). When mice bearing LNCaP tumors reached a tumor volume over 200 mm³ or serum PSA levels reached a minimum of 50 ng/ml, castration was performed via the scrotum under isoflurane anesthesia. Treatment commenced when PSA recovered to pre-castration levels. Mice were randomized into four groups for treatment with SEMA3C ASO or Scr with or without ENZ. For ASO treatment, 12.5 mg/kg of Scr ASO or SEMA3C ASO was intraperitoneally (i.p.) injected once daily for 7 days followed by three weekly treatments thereafter. For ENZ treatment, 10 mg/kg ENZ or

## The paper explained

### Problem

AR pathway-targeted therapies provide the greatest clinical benefit for men with castration-resistant prostate cancer by relieving symptoms and improving survival. However, survival benefit is limited by rapid emergence of lethal treatment-resistant disease within months of therapy. Identifying and targeting critical mechanisms governing treatment resistance represent the next major challenge in prostate cancer research and may lead to development of promising new therapies that have potential transformative impact on clinical management of prostate cancer with near-term clinical benefits.

### Results

Here, we show that SEMA3C is a secreted soluble autocrine growth factor that drives growth and treatment resistance of prostate cancer via activation of multiple RTKs such as EGFR, MET, and ErbB2 in a cognate ligand-independent manner. Furthermore, we show that Plexin B1 is a key SEMA3C receptor in cancer cells. Moreover, Plexin B1 is associated with EGFR and Plexin B1 signaling can drive activation of EGFR. These findings have led us to develop a novel receptor decoy protein inhibitor comprised of Plexin B1 sema domain fused to immunoglobulin Fc domain called B1SP. Herein, we show that this therapeutic protein potently inhibits castration-resistant progression of LNCaP xenografts *in vivo*. Importantly, B1SP is capable of inhibiting RTK pathway activation not only by SEMA3C but also by their cognate ligands, EGF and HGF, making this therapeutic protein an attractive multi-RTK pathway inhibitor targeting MET, ErbB2, and EGFR which are clinically validated targets in a variety of cancers.

### Impact

These studies provide preclinical proof of concept for utility of Plexin B1 receptor decoy protein as a therapeutic protein inhibitor for treatment of advanced prostate cancer. B1SP is a multi-RTK pathway inhibitor targeting EGFR, MET, and HER2/ErbB2 and has potential clinical utility for a number of cancers wherein these RTKs are clinically validated targets including lung, breast, and colorectal cancers. Furthermore, we have found that SEMA3C is a secreted growth factor that can transactivate EGFR and MET signaling in other cancers such as renal, bladder, as well as glioblastoma, thus implicating SEMA3C as a potentially important driver of growth and survival for a broad spectrum of cancers in addition to prostate cancer.

vehicle (5% of Polysorbate 80, 0.9% of benzyl alcohol, and 1% of methyl cellulose) was intragastrically (i.g.) administered once daily 5 days per week. For LNCaP CRPC xenografts, mice were randomized into two groups for treatment with B1SP (20 mg/kg) or PBS control. B1SP or PBS was injected intraperitoneally on two sites/mouse. After treatment start, body weight, tumor size, and serum PSA were continuously monitored weekly. Each experimental group consisted of seven mice. Sample sizes were determined according to our previous *in vivo* assays (Zhang *et al*, 2014). All animal procedures were performed according to the guidelines of the Canadian Council on Animal Care and with appropriate institutional certification.

### Study approval

Animal experiments detailed within the manuscript were approved by the UBC Animal Care Committee, conforming to the mandatory guidelines of the Canadian Council on Animal Care. UBC animal protocol number A15-0150. The study protocol was approved by

University of British Columbia (UBC) Clinical Research Ethics Board (CREB) and Vancouver Coast Hospital Research Institute Research (VCHRI) Research Ethics Board (REB). All patients gave informed consent as approved by UBC CREB and VCHRI REB.

## Statistics

The *in vitro* data were assessed using Student's *t*-test, ANOVA, and the Mann–Whitney test. Data are expressed as the mean ± standard error of the mean (SEM). Comparisons between two means were performed using a Student's *t*-test. Comparisons among multiple means were performed with a one-way ANOVA followed by Fisher' protected least significant difference test (StatView 512, Brain Power, Inc., Calabasas, CA, USA). ANOVA followed by Holm–Sidak *post hoc* analysis was used to compare cell growth. GraphPad Prism software was used to calculate the statistical significance. The threshold of statistical significance was set at $*P < 0.05$, $**P < 0.01$, $***P < 0.001$, and $****P < 0.0001$. Exact *P*-values are indicated in the figure legends, when applicable.

Additional information can be found in the Appendix Materials and Methods section.

**Expanded View** for this article is available online.

## Acknowledgements

We would like to thank Mingshu Dong for producing the graphical abstract and cover art. This work was funded by grants from Prostate Cancer Canada (Grants # TAG2014-06 and GS2015-06), Michael Smith Foundation for Health Research (Grant # 17318), Terry Fox Foundation (Grant # TFF-116129), Canadian Cancer Society Research Institute (Grant # 700347), Cancer Research Society (Grant # F09-60564), the NIH Pacific Northwest Prostate SPORE (NCI P50 CA097186), Prostate Cancer Foundation British Columbia (Grant # F13-0016), and National Centres of Excellence of Canada (CECR PC-TRIADD).

## Author contributions

JWP, AT, NH, KJT, MS, and CJO designed the experiments; JWP, AT, NH, KJT, NAK, NK, MS, DT, LL, WCWL, KCKL, DHFH, HK, LI, TT, TD, PY, IZFJ, SK, and LF performed the experiments; JWP, AT, NH, KJT, MS, AL-FM, LF, AZ, MEG, and CJO interpreted the results; JWP, AT, NH, KJT, MS, MD, AL-FM, AZ, MEG, and CJO wrote the manuscript.

## Conflict of interest

The authors declare that they have no conflict of interest. A patent has been filed pertaining to the results presented in the paper (#WO 2010/ 111792 A1).

## For more information

TCGA, http://cancergenome.nih.gov/.
Chen *et al* (2004), http://www.ncbi.nlm.nih.gov/geoprofiles?term=GDS535+ AND+376_at.

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

I-kappaB degradation to enhance NF-kappaB activity in prostate cancer cells. *Mol Cancer Res* 8: 119−130

