## [Review Process File · EMBO Molecular Medicine]

SEMA3C Drives Cancer Growth by Transactivating Multiple Receptor Tyrosine Kinases via Plexin B1

James W. Peacock, Ario Takeuchi, Norihiro Hayashi, Liangliang Liu, Kevin J. Tam, Nader Al Nakouzi, Nastaran Khazamipour, Tabitha Tombe, Takashi Dejima, Kevin C. K. Lee, Masaki Shiota, Daksh Thaper, Wilson C. W. Lee, Daniel H. F. Hui, Hidetoshi Kuruma, Larissa Ivanova, Parvin Yenki, Ivy Z. F. Jiao, Shahram Khosravi, Alice L-F. Mui, Ladan Fazli, Amina Zoubeydi, Mads Dugaard, Martin E. Gleave, and Christopher J. Ong

Review timeline:

Submission date:	14 February 2017
Editorial Decision:	31 March 2017
Additional correspondence (author)	03 July 2017
Additional correspondence (editor)	17 July 2017
Revision received:	12 October 2017
Editorial Decision:	03 November 2017
Revision received:	22 November 2017
Accepted:	01 December 2017

Editors Roberto Buccione and Céline Carret

Transaction Report:

1st Editorial Decision

31 March 2017

Thank you for the submission of your manuscript to EMBO Molecular Medicine. We are very sorry that it has taken so long to get back to you on your manuscript. In this case we experienced some difficulties in securing three willing and appropriate reviewers.

As you will see, the three Reviewers find the study of interest and worthy of publication. However, reviewer 1 and especially 2, raise a number of concerns. These include reservations on appropriateness of the cell models employed, insufficient experimental support and/or quality for some of the claims, the need to clarify a number of poorly defined aspects and finally, insufficient discussion/mention of prior reports. Reviewer 3, while more positive, also mentions similarly to reviewer 2, the lack of effect of B1SP on the growth of C4-2 CRPC cells in vivo.

During our reviewer cross-commenting exercise, reviewer 1 expressed agreement with the others, while showing less concern about the cell models and the point on B1SP. S/he did stress however, that the specificity of action of the decoy molecules needs to be firmly established.

In conclusion, while publication of the paper cannot be considered at this stage, we would be pleased to consider a revised submission, with the understanding that the Reviewers' concerns must be fully addressed, including the concern about the use of DU145 cells as a model for CRPC, including with additional experimental data where appropriate and that acceptance of the manuscript will entail a second round of review.

Provision of the author checklist is mandatory at revision stage. In this case, the author checklist is especially relevant as, in addition to the concerns on the clinical features of the TMA, I note that both reviewers have reservations on your presentation of statistics information.

I look forward to seeing a revised form of your manuscript as soon as possible.

Additional correspondence (author)

03 July 2017

We are writing to update you on the status of our revision as of the 3 month anniversary date of the decision letter. As of today, our revision is almost complete. We now have all the in vitro data addressing reviewers comments. The only outstanding experiment left is to perform an in vivo C4-2 experiment to address the reviewers concerns that the in vivo C4-2 data presented in the manuscript did not show a very strong inhibition with B1SP treatment. The reason for this is that C4-2 are fast growing and highly aggressive tumors and we had to terminate the experiments early as they reached humane endpoints set by our ethics protocols. We have repeated the experiment by inoculating fewer cells (1 million cells versus 2 million cells) and only one tumour per mouse so that we can effectively treat with B1SP for a longer period of time. This should give us better separation between B1SP treated and control groups. The C4-2 in vivo experiment is currently underway and the tumors have just started to come up. We began treating mice with B1SP this week. If we need to include this in vivo data, we will need to ask for an extension of an additional 8-10 weeks. At this time, we would like to ask for your guidance on whether you think the in vivo C4-2 data is necessary for the revision. This data is only included as part of the supplementary data and the main conclusion of the paper is that B1SP significantly delays growth of LNCaP xenografts post castration as shown in Figure 8. If the C4-2 data is not necessary then at this point, we have addressed all other concerns raised by the reviewers and will be able to submit a substantially revised manuscript within 1-2 weeks. We look forward to your recommendation.

Additional correspondence (editor)

17 July 2017

I went through the reviewer evaluations again. Although I cannot presume to reply on their behalf, both #1 and 2 mentioned the need for better C4-2 data. However, the point you make is reasonable.

In conclusion therefore, while I would have no objection to extending your revision period further as necessary, it is ultimately up to you to decide as to whether to proceed now. Should you decide for the latter, it is my recommendation to explain in detail (as below) to the reviewers the reasons for not performing the experiment.

1st Revision - authors' response

12 October 2017

Point-by-point response to Referees' comments:

Referee 1

In the current study the authors examined the role of SEMA3C/Plexin B1 axis in prostate cancer growth and castration-resistant progression. They first demonstrated increased expression of SEMA3C in prostate cancer cell lines and patient samples. They further demonstrated that the action of SEMA3C was largely mediated through semaphorin receptor Plexin B1-dependent activation of RTKs. By using both gain and loss of function approaches, they showed that SEMA3C promotes prostate cancer cell growth and survival by promoting cell cycle progression and inhibiting apoptosis. The authors also nicely showed that treatment of antisense oligo of SEMA3C inhibits growth of hormone-sensitive and castration-resistant prostate tumors in mice. Finally, they engineered decoy molecules to inhibit the interaction between SEMA3C and Plexin B1 and demonstrated that anti-tumor efficacy of these molecules. Overall, the studies are well designed and carried out

and very systematic with many innovative discoveries. The conclusion is generally supported by the data obtained. However, a number of issues remain to be addressed.

Figure 1:

SEMA3C ligand is a secreted protein. To prove the cell surface staining of the antibody it is important to use one of the cell lines examined in Figure 1C to perform SEMA3C KD experiments and perform SEMA3C IFC staining to verify the staining pattern of SEMA3C seen in patient samples are real.

As suggested by the reviewer, we knocked down SEMA3C in DU145 cells using our siSEMA3C-1. siScramble (siScr) was used as control. The cells were then fixed and stained for SEMA3C using the same SEMA3C (N20), Santa Cruz) antibodies as employed in Figure 1A. Secondary antibodies were used as control. We observed decreased SEMA3C staining in siSEMA3C treated cells compared to the siScr control. This data supports the specificity of SEMA3C staining in our IHC patient samples presented in Figure 1A. We have added this data to the expanded view, Figure section (Figure S1A).

Figure S1. High SEMA3C expression is associated with castrate resistant tumors. (A) Confocal images of SEMA3C expressed in DU145 cells transfected with siRNA scramble (siScr) or siSEMA3C. DU145 cells were stained with secondary antibodies alone (left panel) or SEMA3C (N20) antibodies. Knock down of SEMA3C protein levels is shown in the accompanying immunoblot probed with SEMA3C (N20) antibodies and actin for loading control.

Also, the cBioPortal data shown in Figure 1E are interesting, but can the authors verify the data in the samples used in Figure 1A.

We agree with reviewer that verifying the cBioportal data in clinical tissue samples used in Fig 1A would provide valuable information regarding activation of signalling pathways within tumour samples and may have potential implications for prognosis and treatment. However, unfortunately, the procurement procedure used for the formalin fixed tissue specimens in Fig 1A was not designed to specifically preserve protein phosphorylation. While formalin fixation preserves tissue histomorphology, formalin penetrates tissue slowly and is unsuitable for stabilizing highly labile phospho-epitopes. Furthermore, for phosphoprotein analyses by IHC, ischaemia needs to be avoided during tissue collection since ischaemia influences protein phosphorylation in a tumour specific and unpredictable manner. Finally, we

are unable to obtain any additional sections of TMA for these studies since the number of tissue sections of the TMA containing the metastatic CRPC samples from the UW rapid autopsy program is highly limited and restricted. Because of the limitations noted above, there is uncertainty over whether staining levels can be interpreted relative to its relationship with actual phosphorylation states in vivo.

Figure 2:

The description of PLA assay in figure legend and main text is not clear. How would the readers know the signal/staining shown in Figure 2B indicates the engagement of SEMA3C with Plexin B1?

We have clarified the first introduction to PLA interactions in the legend of Figure 2B to read “Association of SEMA3C and PLEXINB1 (interactions/cell)”. We have kept the Y-axis labelling identical in all PLA figures such that, for example, “SEMA3C-NRP1 (interactions/cell)” should be interpreted as the association between SEMA3C and NRP1 (interactions/cell) etc.

In this case, it is also critical to include PLA images (both SEMA3C-Plexin B1 and SEMA3C-NRP1 interactions) in Plexin B1 KD cells.

As suggested by the reviewer, we have performed PLA analysis between SEMA3C and PLEXINB1 in si-PLEXINB1 KD DU145 cells treated with SEMA3C:Fc. This data now appears in Figure 2C with corresponding photomicrographs. We repeated the data shown in the previous version (Figure 2C) that demonstrated the association between SEMA3C and NRP1 in DU145 siPLEXINB1 KD cells and have included the corresponding representative photomicrographs. This replacement data now appears as Figure 2D.

Fig. 2. SEMA3C acts through Plexin B1. (C) PLA interactions of SEMA3C and NRP1 in control (rhIgG₁Fc) or SEMA3C:Fc-treated DU145 cells transfected with either siPLXNB1 or scrambled siRNA (siScr). (D) PLA interactions of SEMA3C and NRP1 in control (rhIgG₁Fc) or SEMA3C:Fc-treated DU145 cells transfected with either siPLXNB1 or si-Scr.

Figure 3:

The WB data in parental LNCaP cells shown in Figure 3A is not consistent with the data shown in Figure 1C and needs to be repeated.

We thank the referee for pointing this out and we apologise for the confusion. The experiment shown in Figure 3A compares growth of LNCaP overexpressing full length SEMA3C (LNCaP_{SEMA3C-FL}) versus empty vector transduced LNCaP cells as control (LNCaP_{empty}). The description in the Figure legend for the experiment shown in Figure 3A, (“Inset shows immunoblot of SEMA3C protein levels in LNCaP_{empty} and LNCaP_{SEMA3C-FL} conditioned medium”) was in error and was an immunoblot of whole cell extracts isolated from LNCaP_{empty} or LNCaP_{SEMA3CFL} cells not conditioned medium as described in the figure legend. The Western blot in the former version of the paper was a HIS tag blot showing the exogenous SEMA3C expression that was labelled SEMA3C. We have made that clarification in the Figure legend (“Inset shows immunoblot of SEMA3C protein levels in LNCaP_{empty} and LNCaP_{SEMA3C-FL} using SEMA3C (N20) and HIS-tag antibodies”) and we have added the data that shows the SEMA3C expression levels from cell lysates of LNCaP_{empty} and LNCaP_{SEMA3C-FL} cells. The immunoblot shows the endogenous SEMA3C levels of the LNCaP_{empty} cells that was detectable but low and high levels of SEMA3C in LNCaP_{SEMA3C-FL} cells. We included immunoblots probed with both SEMA3C-specific (N20) and HIS antibodies to demonstrate the overexpression of our HIS-tagged SEMA3C construct. The blot was re-probed with actin antibodies as loading control. This new data appears in Figure 3A.

Fig. 3. SEMA3C regulates prostate cancer cell growth. (A) Growth of LNCaP_{empty} and LNCaP_{SEMA3C-FL} cells cultured in medium containing 5% fetal bovine serum (FBS) as monitored by cell counting. Inset shows immunoblot of SEMA3C protein levels in LNCaP_{empty} and LNCaP_{SEMA3C-FL} using anti-SEMA3C (N20) and anti HIS-tag antibodies.

The cell growth experiments shown in Figure 3B need to be repeated since the same number cells should be used for untreated cells.

The cell growth data shown in Figure 3B represents cell number at 72 hours following plating of equal number of cells and culturing in growth conditions containing varying concentrations of R1881. The “0” concentration point in Figure

3B represents the cell number of the LNCaP_{empty} cells compared to LNCaP_{SEMA3C-FL} cells after 72 hours of growth in the absence of R1881 treatment. Treatment of LNCaP_{SEMA3C} cells therefore showed a dose-dependent increase in cell growth on day three following synthetic androgen treatment. We have changed the wording in the results section to clarify the experiment. We also changed the data format to a histogram plot more traditionally used for single time point data sets.

“SEMA3C overexpression also led to increased cell growth in androgen-free serum (CSS) conditions following 72 h of culture (Figure 3B) and an augmented growth response to increasing concentrations of synthetic androgen R1881 (Figure 3B).”

Fig. 3. SEMA3C regulates prostate cancer cell growth. (B) Growth of LNCaP_{empty} and LNCaP_{SEMA3C-FL} cells treated with R1881 (0-0.5nM) was assessed after 72 hrs as above.

Figures 4 and 5:

The evidence for the autocrine effect of SEMA3C for the data seen in Figure 4 is very weak currently. In other words, the authors did not rule out the possibility that SEMA3C KD may affect RTK phosphorylation in a manner independent of its ligand function.

To demonstrate that secreted SEMA3C can drive RTK phosphorylation, we performed an experiment whereby conditioned medium was harvested from stable HEK293T cells overexpressing a HIS-tagged full-length SEMA3C or empty vector transfected HEK293T cells as control. The cells were cultured for 48h in serum-free medium, CM was harvested and concentrated 8-fold. The conditioned medium was then applied to LNCaP cells that were serum starved for 24h as a mix of HEK293T_{SEMA3CFL} CM and HEK293T_{empty} CM ranging from 100% to 12.5% HEK293T_{SEMA3CFL} CM for 20 min. Recombinant SEMA3C-Fc protein was applied as a positive control. We then harvested whole cell lysates for immunoblotting. The data show a dose-dependent reduction of EGFR, SHC, MAPK phosphorylation levels with HEK293T_{SEMA3CFL} CM dilution. Vinculin levels are shown as a control for loading. Moreover, EGFR, SHC, MAPK phosphorylation levels are attenuated by immunodepletion of SEMA3C from HEK293T_{SEMA3CFL} CM.

This new data taken together with the SEMA3C KD data in Figure 4A and the SEMA3C KD rescue data in Figure 4C, and 4I showing that exogenous SEMA3C can restore RTK pathway activation and proliferation in SEMA3C KD cells and the SEMA3C induced growth is mediated via plexin B1 as shown in Figure 2A collectively support the notion that SEMA3C drives RTK pathway activation and

proliferation in a ligand-dependent, autocrine manner. The new data has been added to the results section under subhead 2 and appears in the expanded view as Figure S1D in the manuscript.

Subhead 2 “To investigate whether naturally secreted SEMA3C could activate the EGFR/ErbB2 signaling pathway, we treated LNCaP cells with conditioned medium harvested from HEK 293T cells that over express full-length SEMA3C. We observed a dosage-dependent activation of EGFR, SHC and MAPK phosphorylation with increasing concentration of SEMA3C conditioned medium suggesting that SEMA3C is an autocrine growth factor that acts upstream of EGFR signaling (Fig. S1D).”

Figure S1 (D) LNCaP cells treated with either recombinant SEMA3C-Fc fusion protein (0.5 μ M), or conditioned medium (CM) from HEK 293 T cells alone (0) or, mixed in the indicated proportion with conditioned medium from HEK 293T that stably overexpress and secrete natural Full-length SEMA3C. The immunoblot shows the levels of EGFR, SHC and MAPK phosphorylation after 20 minutes of treatment. Vinculin levels are shown as loading control (upper panel). The lower panel shows SEMA3C levels and EGFR, SHC and MAPK phosphorylation levels in DU145 cells treated with CM as above or SEMA3C immunodepleted using anti- SEMA3C N20 (2 μ g/ml) CM. Vinculin levels are shown for loading control.

It is unclear why the authors need to skip the dose of 0 nM of ASOs in experiments shown in Figure 4G.

The data in Figure 4G was expressed as a percentage of maximum growth for the respective treatment (ie. oligofectamine alone). The maximum growth was achieved in the absence of ASO treatment in each case and is therefore 100%. In order to reduce confusion we have added the data to the figure.

It is also unclear why the authors focused on the effect of SEMA3C OE on cell cycle but on the effect of ASOs on apoptosis, which is not entirely in line with the experiments done in mice shown in Figure 5.

It is common for growth factors and potential oncogenes to drive the cell cycle that culminates in enhanced cell growth. Likewise, specific knock down or inhibition of oncoprotein expression often results in increased cellular apoptosis. LNCaP cells are mutant for PTEN and as such under normal growth conditions only shows minimal apoptosis. Our data as shown in Figure 3C demonstrates that the overexpression of SEMA3C under normal growth conditions can drive the cell cycle, a possible mechanism for the observed enhanced cell growth. The data in Figure 4 G, J and K demonstrate that the treatment of C4-2 cells with SEMA3C ASO compared to Scramble control resulted in a dose-dependent inhibition of cell growth. SEMA3C ASO-treated cells also demonstrated a dose-dependent increase of apoptosis as measured by an increased sub G0/G1 DNA content, PARP and caspase-3 cleavage.

We cultured LNCaP cells under serum deprivation conditions (0.5% CSS) and cultured them for 4 days in the absence (PBS) or presence of SEMA3C. The cells were then harvested and assayed for subG0/G1 DNA content using propidium iodide staining and flow cytometry. We observed a significant (30%) decrease in sub G0/G1 DNA content of LNCaP cells treated with SEMA3C compared to PBS control under these growth conditions further demonstrating that SEMA3C overexpression may promote cell growth in-part by inhibiting cellular apoptosis. This additional data has been added to the manuscript and presented in Figure 3E. SEMA3C antisense treatment in LNCaP xenograft mice showed reduced cell proliferation by decreased levels of Ki-67 staining and increased TUNEL IHC staining of tumor tissue sections derived from Xenograft tumors compared to controls further supporting the notion that the inhibition of SEMA3C results in increased apoptosis (Figure 5 C and G). We have removed Figure 3E of the previous version to be more consistent with the theme that inhibition or overexpression of SEMA3C can have a role in cellular apoptosis.

Figure 3 (E) LNCaP cells (2.5×10^5)/well were treated in RPMI supplemented with 0.5% CSS in the absence (PBS) or presence of SEMA3C:Fc ($0.5 \mu\text{M}$) for 4 days. Cells were then harvested for propidium iodide staining and analyzed for apoptosis by flow cytometry for the proportion of cells in sub G₀/G₁ DNA content of the cell cycle. Bars represent the Mean and SEM percent of maximum apoptosis relative to control; treatments performed in triplicate ($p=0.016$).

The quality of IHC images shown in Figure 5G should be improved.

We have improved the quality of these images.

Figure 6:
Statistical analyses are missing in the animal studies shown in Figure 6.

We have added the statistical data for this experiment (Figure 6E and F). Statistical significance was reached 2 weeks after treatment. We have added the statistical significance to the Figure legend.

Figures 7 and 8:
In studies in these figures the authors engineered decoy protein to functionally disrupt SEMA3C-induced RTK activation. However, it is important to include a deficient mutant as negative control to demonstrate the specificity of the action of these decoy molecules.

To demonstrate the specificity of B1SP decoy protein, we have generated an analogous Plexin D1:Fc fusion protein containing the SEMA domain and adjacent PSI domain of Plexin D1 fused to human IgG1Fc1 (D1SP) as a control since our data as shown in Figure 2A demonstrated that siRNA knock down of Plexin B1 but not Plexin D1 was able to inhibit LNCaP cell growth. The schematic of D1SP:Fc appears as Figure S5B. To demonstrate the specificity of our B1SP protein we performed proliferation and signaling assays in LNCaP cells as shown in Figure 8A. We have added a comparison of proliferation with LNCaP cells treated with increasing concentration of purified B1SP versus D1SP recombinant Fc-fusion proteins. B1SP treatment inhibited cell growth of LNCaP cells by approximately 80% at a greatly reduced dosage compared to D1SP (Figure S5C). To compliment

this data we also examined SEMA3C induced signaling in LNCaP and DU145 cells treated with D1SP. There was no change with increasing D1SP treatment in EGFR, Her2/ErbB2 and downstream phosphorylation in LNCaP cells. There was also no inhibitory effect of D1SP treatment on EGF and HER2/ErB2, nor MET phosphorylation and downstream signaling in DU145 cells. We have added this data to Figure S5E and F. Whereas, the treatment of LNCaP cells with B1SP inhibits SEMA3C-mediated phosphorylation of the EGFR and downstream signaling as shown in Figure 8E, D1SP treatment under the same conditions had no effect on EGFR, Her2/ErbB2, SHC and MAPK phosphorylation. The data suggests that the inhibition of the SEMA3C-PlexinB1 signaling with B1SP treatment but not Plexin D1 via D1SP treatment specifically inhibits RTK pathway activation in prostate cancer cells.

Figure S5C

Fig. S5. Plexin B1 Decoy protein inhibits Cell Growth, Semaphorin and RTK signaling and Tumor growth *in vivo*. (C) Cell growth of LNCaP cells treated with either B1SP or D1SP at the indicated concentrations for four days. Cell growth was assayed using the Prestoblué cell proliferation reagent. Data represents the Mean and SEM of triplicate wells. The data is representative of three independent experiments. ** $p < 0.01$

E

F

Fig S5 (E) LNCaP (3×10^5 /well) cells were serum starved for 24h treated with PBS or D1SP as indicated in the absence of serum for 1 h followed by stimulation with SEMA3C(0.5 μ M) for 20 min. Cell lysates were run on SDS-PAGE for Western Blot analysis. Immunoblots were probed with EGFR, Her2/ErbB2, SHC and MAPK phospho-specific antibodies as shown. Blots were re-probed with EGFR, Her2/ERbB2, SHC and MAPK antibodies and vinculin for loading controls. The data is representative of two independent experiments. (F) DU145 (3×10^5 /well) were serum starved in the absence or presence of D1SP at the indicated concentration for 3 hr. The media was then changed and replaced with D1SP containing media in the absence or presence of SEMA3C (0.5 μ M) for 20 minutes. Protein cell lysates were separated on SDS-PAGE for Western Blot analysis. Immunoblots were probed with EGFR, Her2/ErbB2, MET, Gab-1, SHC, AKT and MAPK phospho-specific antibodies as shown. The Blots were re-probed with EGFR, Her2/ERbB2, MET, Gab-1 SHC, AKT and MAPK antibodies and vinculin for loading controls. The blot is representative of three independent experiments.

Referee #2 (Comments on Novelty/Model System):

Whereas there are significant novel findings in this paper, some observations have been made in other systems. The inhibitors that have been developed, at their present state, are unlikely to have medical value. Additional model systems for some experiments are warranted. Please see the remarks to the authors for details.

Referee #2 (Remarks):

In this manuscript, Peacock et al. report that SEMA3C expression is associated with castration resistant prostate cancer (CRPC) and that it promotes PC cell growth in vitro and in vivo. Evidence is provided showing that SEMA3C functions via PLEXIN B1 and NRP1/2 to activate multiple receptor tyrosine kinase signaling pathways in PC cells. In vivo experiments show that SEMA3C over-expression promotes orthotopic and castration resistant growth of PC xenografts, while inhibition of SEMA3C reverses these effects in preclinical models. The data implicate SEMA3C signaling as a potential therapeutic target for PC, in particular CRPC.

This is a well-written manuscript with significant data consistent with the conclusions drawn. However, there are a number of points that need to be addressed:

1. SEMA3C is a secreted protein. The authors use recombinant SEMA3C:Fc fusion in various experiments to study the effect of secreted SEMA3C. It is important to determine if the naturally secreted form of SEMA3C has similar activity. This could, for example, be done by using conditioned medium from cells, in which either SEMA3C is knocked down or overexpressed, on other cell types.

To test whether the naturally secreted form of SEMA3C is able to drive RTK pathway activation, we performed an experiment as suggested by the reviewer, whereby conditioned medium (CM) was harvested from stable HEK293T cells overexpressing a HIS-tagged wildtype full-length SEMA3C. The cells were cultured for 48h in serum-free medium, CM was harvested and concentrated 8-fold. The HEK293T_{SEMA3CFL} conditioned medium was then applied to LNCaP cells that were serum starved for 24 hr as a mix of serial 2-fold diluted with HEK293T_{empty} CM from 100% HEK293T_{SEMA3CFL} to 12.5% HEK293T_{SEMA3CFL} containing CM for 20 min. Recombinant SEMA3C-Fc protein was applied as a positive control. We then harvested whole cell lysates for immunoblotting. The data show a dose-dependent reduction of EGFR, SHC, MAPK phosphorylation levels with HEK293T_{SEMA3CFL} dilution. Vinculin levels are shown as a control for loading. Moreover, EGFR, SHC, MAPK phosphorylation levels are attenuated by immunodepletion of SEMA3C from HEK293T_{SEMA3CFL} CM. This new data taken together with the SEMA3C KD rescue data in Figures 4C and 4I showing that exogenously added SEMA3C can restore proliferation and RTK signalling in SEMA3C KD cells demonstrate that SEMA3C can activate EGFR signaling in a ligand-dependent, autocrine manner. This new data has been added to the results section under subhead 2 and appears in the expanded view as Figure S1D in the manuscript. The data is shown below.

Subhead 2: “To investigate whether naturally secreted SEMA3C could activate the EGFR/ErbB2 signaling pathway, we treated LNCaP cells with conditioned medium (CM) harvested from HEK 293T cells that over express full-length wild-type SEMA3C. We observed a dosage-dependent increase in EGFR, SHC and MAPK phosphorylation with increasing concentration of SEMA3C containing CM and a corresponding decrease in EGFR, SHC and MAPK phosphorylation in SEMA3C immuno-depleted CM, suggesting that SEMA3C is an autocrine growth factor that drives EGFR activation (Figure S1D).”

(D) LNCaP cells treated with either recombinant SEMA3C-Fc fusion protein (0.5 μ M), or conditioned medium from HEK 293 T cells alone (0) or, mixed in the indicated proportion with conditioned medium (CM) from HEK 293T that stably overexpress and secrete natural Full-length SEMA3C. The immunoblot shows the levels of EGFR, SHC and MAPK phosphorylation after 20 minutes of treatment. Vinculin levels are shown as loading control (left panel). The right panel shows SEMA3C levels and EGFR, SHC and MAPK phosphorylation levels in DU145 cells treated with CM as above or SEMA3C immuno-depleted using anti-SEMA3C N20 (2 μ g/ml) CM for 20 min. Vinculin levels are shown for loading control.

In addition, SEMA3C is known to be cleaved and the different cleavage products have been suggested to have different, sometimes opposing, functions. Which SEMA3C products are the predominant forms in different PC cell models?

We have observed all forms of SEMA3C secreted from all of the prostate cancer cell lines. There is typically a doublet that runs at about 89 kDa that likely represents the full-length protein and the c-terminal cleavage product often referred to as the Δ 13 respectively. We also typically see a band that runs below 70kDa that represents the p65 cleavage product. Below we show an expanded view of

SEMA3C in the prostate cancer cell lines. Higher levels of full length SEMA3C are secreted from CRPC, PCa lines C4-2, DU145 and 22Rv1 compared to LNCaP and LNCaP-derived, ENZ resistant MR49F cells. All Prostate Cancer cell lines secrete the p65 SEMA3C at levels equivalent or higher than LNCaP. In general, the level of p65 protein secreted may correlate with the amount of full-length protein secreted but there is more SEMA3C p65 present than full length SEMA3C in all cell lines with the exception of DU145 which has apparent equal distribution of the full length and the p65 cleavage product (Figure 8 below).

SEMA3C expression in prostate cancer cell lines. Cells were seeded at 2×10^5 /ml in 6-well tissue culture plates. Twenty-four hours later the medium was changed to serum-free and the cells were cultured for an additional 48 hr. CM was then harvested and an equivalent volume of CM from the PCa lines was run on SDS-PAGE and immunoblotted using SEMA3C antibodies. The blot shows the p89 and p65 forms of SEMA3C. The blot was stained with ponceau to show equal protein levels in the CM.

2. On page 12, authors state "Under castrate conditions, LNCaP_{SEMA3C-FL} tumors exhibited enhanced tumor growth rates as compared to LNCaP_{empty} controls, implying that SEMA3C promotes castration resistant growth (Figure 3F)." These data need to be supported by experiments in CRPC cell lines. Please note that the use of the inhibitor B1SP resulted in a modest difference in tumor growth at a single time point (Fig. S5H).

We appreciate the referee's concern that the tumor volume of C4-2 xenografts shown in Figure S5H reached statistical significance at only the experimental endpoint. The experiment shown was done by injecting 2 million cells per site and two sites per mouse. C4-2 cells grow quite aggressively and because of the rapid developing tumor burden the experiment had to be terminated due to tumor burden endpoints as per our animal ethics protocols just as we reached statistical significance at the last time point as shown.

To address this valid concern we repeated the C4-2 xenograft experiment using 1 million cells/site and only one injection site per mouse 5, control (PBS) and 7, B1SP-treated mice. We found that there was a tighter correlation between tumor volume and serum PSA levels when we used one site/mouse. Changes in the serum PSA levels result from a single developing tumor. Inhibition of tumor volume and

serum PSA began to reach statistical significance as of week six in the C4-2 xenograft model. The new data as shown in Figure S6 D and E illustrate clear tumour growth inhibition with B1SP and we have replaced the original figure in the former Figure S5H with this data.

Fig. S6 (D) Tumor volume (mm³) and (E) PSA (ng/ml) of from athymic *nu*^{-/-} mice bearing C42 Tumors treated with either PBS (n=5) or B1SP (n=7) post castration over a period of 10 weeks. Data are representative of Mean +/- SEM, **p*<0.05

3. Fig. 4: As a model for castration resistance, the authors use DU145 cells. Given that these cells do not express the AR, and that SEMA3C is regulated by AR (published by the authors themselves), it is necessary to use other appropriate models, such as C4-2B or 22Rv1 cells that express AR and are CRPC models.

We agree with the referee that DU145 is not a model for AR-driven castration resistance.

The data in Figure 4 demonstrates that autocrine SEMA3C signaling drives prostate cancer cell growth. We chose DU145 cells in this case, as a model of non-AR driven (ie. AR-indifferent CRPC) because they express SEMA3C at a level that we could inhibit by specific knock down using SEMA3C siRNA and study the consequences for RTK signaling and cell growth. The data was intended to:

1. Demonstrate that specific knockdown of SEMA3C inhibits RTK signaling.
2. Show that SEMA3C knock down initiates apoptosis.
3. Prove that inhibition of Plexin B1 a potential receptor for SEMA3C results in inhibition of RTK signaling.
4. To provide evidence that SEMA3C knock down inhibits cell growth that can be rescued by recombinant SEMA3C treatment.

Loss of AR is amongst several mechanisms driving androgen independence. For this set of experiments, we have shown the consequences of SEMA3C knockdown on cell proliferation and RTK signaling, a potential mechanism underpinning the biological activity of SEMA3C in PCa.

We agree with the reviewer that C4-2 and 22Rv1 are appropriate models of AR driven CRPC. Data on C4-2 model for AR-driven CRPC is shown in Figure 8I, J and S6D and E and 22Rv1 data are shown in Figure S4. Further evidence for the involvement of SEMA3C in CRPC is shown in Figure 4E-I. Antisense oligonucleotide knock down of SEMA3C in CRPC, C4-2 cells inhibits cell growth that can be rescued with recombinant SEMA3C treatment (Fig 4G and I), and activates apoptosis (Figure 4J and K). Moreover, the inhibition of SEMA3C signaling using our B1SP inhibitor also inhibits cell growth and RTK signaling in C4-2 and 22RV1 cells (Table S2, Figures 8F and H and S5G).

Fig. S6 **(D)** Tumor volume (mm³) and **(E)** PSA (ng/ml) of from athymic *nu*^{-/-} mice bearing C42 Tumors treated with either PBS (n=5) or B1SP (n=7) post castration over a period of 10 weeks. Data are representative of Mean +/- SEM, **p*<0.05

4. Fig. 4B. Apoptosis should be directly assessed, such as with TUNEL. Another marker of apoptosis could be used, such as cleaved Caspase 3.

As suggested, we have repeated this experiment and added the cleaved caspase-3 data. Figure 4B

Figure 4 (B) DU145 cells transfected with siScr, siSEMA3C-1 or siSEMA3C-2 and cultured in medium containing 10% FBS for 72 hrs. Apoptosis was demonstrated by immunoblotting with cleavage specific-PARP and -caspase-3 antibodies and with PARP antibodies that recognizes both native (116kDa) and cleaved (89kDa) PARP. Vinculin is shown as loading control. The data is representative of three independent experiments.

5. The quality of the western analysis for some antisera could be improved.

We have repeated blotting of the western analyses that we felt we could make improvements to the quality. Specific changes were made in Fig 4 A, B and H.

How many times were the western experiments repeated? This should be indicated in the figure legends.

As requested, we have now indicated in the Figure legends the degree of reproducibility of the experiments. All experiments have been performed independently a minimum of three times unless otherwise stated in the Figure legend.

6. Fig. 6A. Expression in more tumor samples should be shown, esp. for sensitive tumors.

Unfortunately, we do not have any additional tumor samples than what is already shown for this experiment.

Fig. 6E, significance should be indicated.

We have added the statistical significance for the data shown in Figure 6E and in the legend for Figure 6. Statistical significance was reached two weeks after the

initiation of treatment, * $P < 0.05$, ** $P < 0.01$.

7. Fig. 7A. The reasoning for the specific structure of the recombinant decoy proteins are not specified. An uncleavable SEMA3C was previously described as a tool to inhibit SEMA3C; could that be used in the current experiments? This should be discussed.

Yang *et al.* and Mumblat *et al.* both describe SEMA3C constructs that inhibit the function of SEMA3C. In our study, we used uncleavable full-length SEMA3C containing mutations in both furin cleavage sites. Yang *et al.* used a cleavable full-length construct and Mumblat *et al.*, used a construct (FR-SEMA3C) that is truncated at bp 2216 before the third furin and ADAMTS1 site. Toledano *et al.* used the same construct that was described by Mumblat *et al.* We have found that a SEMA3C, c-terminal truncation mutant, ($\Delta 13$), that is analogous to the FR-SEMA3C described by Mumblat *et al.*, behaves as an antagonist that inhibits PCa cell growth and also inhibits EGFR (and downstream) phosphorylation similar to our SEMA3C-SD (SD-ALB) that includes the SEMA3C semadomain and adjacent PSI domain, N-Terminally fused to human Albumin, shown below (Figure 11). We have included a representative blot comparing our SD-ALB (ALS) to $\Delta 13$ with regard to the inhibition of EGF-mediated EGFR phosphorylation (Figure 10). SEMA3C-mediated EGF receptor signaling is inhibited by our ALS inhibitor (see below). We have also included cell signaling data with regard to the inhibition of EGFR, Her-2 signaling and downstream effectors by cognate ligand EGF (Figure 11D). Cellular proliferation of LNCaP (Figure 11B) and 22Rv1 (Figure 12) cells treated with either our ALS or $\Delta 13$ SEMA3C fusion proteins are shown for comparison. Moreover, Lentiviral SEMA3C-SD inhibits orthotopic LNCaP tumor development and serum PSA compared to empty virus *in vivo* (Figure 11E and F). The effect of inhibition of cell proliferation of both these constructs was similar. Owing to the Albumin fusion, the utility of our ALS SEMA3C inhibitor is not an ideal clinical candidate for a potential therapeutic for treatment of PCa. For this reason, we sought to inhibit the SEMA3C-RTK signaling axis by developing an Fc-fusion inhibitor molecule “B1SP” capable of inhibiting the SEMA3C-mediated Plexin B1 signaling axis at the level of the Plexin B1 receptor. The rationale for the design and the development of our B1SP fusion protein as a truncated Plexin B1 decoy that could potentially heterodimerize with the endogenous Plexin B1 receptor and thus potentially inhibit SEMA3C-mediated Plexin B1 RTK transactivation.

C-terminal SEMA3C truncation mutant inhibits EGFR phosphorylation. LNCaP cells were treated with either ALS or c-terminal SEMA3C truncation mutant ($\Delta 13$) for 1 h followed by EGF(10ng/ml) stimulation. Immunoblots were probed with p-EGFR antibodies and then reprobbed with EGFR and Vinculin as loading controls. The blot is representative of several repeated experiments.

Figure 11. Proteolytic cleavage product of SEMA3C inhibits cell growth. (A) Diagram (left) shows sema, PSI and Ig-like domains of full-length SEMA3C homodimer. The arrow indicates the processing consensus sequence-1 (K/RXRR) site. The right graphic shows the monomeric SEMA3C sema domain containing proteolytic cleavage product (SEMA3C-SD) (65 kDa). (Right) Proteolytic processing of SEMA3C was monitored by immunoblot analyses of SEMA3C in CM from full-length SEMA3C expressing HEK 293T cells treated with Furin inhibitor at the indicated doses using SEMA3C N-terminal peptide (N-20) specific Abs. Bar graph shows % SEMA3C-SD in CM. Relative amounts of SEMA3C FL and SEMA3C-SD were determined by densitometric analyses. % SEMA3C-SD = SEMA3C-SD/(SEMA3C FL +

SEMA3C-SD) \times 100%. **(B)** Growth of LNCaP cells treated with SEMA3C:Fc (0.5 μ M) +/- SD-ALB (2.0 μ M). LNCaP cells treated with PBS were used as control. Similarly, LNCaP cells were treated 4 days with EtOH vehicle alone (control), or R1881 (1 nM) +/- SD-ALB (2.0 μ M), (lower panel) and growth was measured using Presto Blue assay (mean +/- SEM, n=3). **(C)** LNCaP cells were serum starved and treated with either albumin (2.0 μ M) or with SD-ALB (0-2.0 μ M) for 60 min followed by stimulation with SEMA3C:Fc (0.5 μ M) for 10 min. Levels of total and phosphorylated EGFR and SHC were determined by immunoblot analyses. **(D)** Serum starved LNCaP cells were treated with albumin (2.0 μ M) or SD-ALB (0-2.0 μ M) for 60 min. Followed by stimulation with EGF (10 ng/ml) for 10 min. Cells were then washed and lysed. The effect of SD-ALB treatment on EGF, HER2/ErbB2, SRC, SHC and MAPK phosphorylation was determined by immunoblotting using phospho-specific and total Abs. **(E)** Tumor volume (mm^3) and **(F)** PSA (ng/ml) from athymic *nu*^{-/-} mice bearing LNCaP tumor treated with high titre S3C-SD lentivirus or empty vector as control post-castration over a period of 10 weeks. Data represent mean +/- SEM, n=10.

Figure 12. C-terminal SEMA3C truncation mutant Δ 13 inhibits cell proliferation. 22Rv1(3,000/well) were seeded on 96-well plates in triplicate. Twenty-four hours later the growth medium was replaced (RPMI + 0.2%CSS) containing either PBS as control or ALS or Δ 13 (2.0 μ M). Cell growth was assessed initially and 48 hours after treatment initiation using the Presto Blue proliferation assay.

8. Figs. 8I and G: Does B1SP reach the tumors and affect the pathways in vivo similar to as found in vitro? Data should be shown.

To address this point, we harvested LNCaP xenograft tumors from mice treated for two weeks with PBS or B1SP. Tumors were harvested one hour after the final injection of B1SP or PBS as control. Tumors were divided in half; tissue sections were prepared for Immunohistochemistry and tissue cell homogenates were lysed for protein analysis by Western blotting. The data are shown in Figure S6 A-C. Using IHC, the presence of B1SP was detected in LNCaP xenograft tumor sections using antibodies against the HIS-tagged B1SP protein as well as reduced levels of EGFR and MAPK phosphorylation levels in B1SP-treated tumor tissue (Figure S6A). Moreover, we were able to demonstrate reduced phosphorylation levels of MAPK from protein lysates from tumor tissue derived from B1SP-treated mice (Figure S6B) and reduced cellularity and proliferation by H&E and Ki-67 staining

respectively (Figure S6C). These data demonstrate B1SP is able to reach the tumors, and provide pharmacodynamic evidence of target pathway inhibition.

Fig. S6. B1SP is delivered to LNCaP xenograft tumors in vivo. (A) IHC staining of tissue sections derived from LNCaP xenograft tumors isolated from PBS and B1SP-treated mice. B1SP was detected by staining with HIS antibodies, the phosphorylation levels of MAPK are shown. (B) Immunoblot showing the levels of MAPK phosphorylation in PBS and B1SP treated xenograft tumors is shown (C) IHC showing cellularity (H&E) and proliferation (Ki-67) of tissue sections derived from PBS and B1SP treated mice bearing LNCaP xenograft tumors.

9. SEMA3C has been implicated in angiogenesis and lymphangiogenesis, with some conflicting results in the literature in different tissues. Does SEMA3C inhibition in vivo in the pre-clinical PC models affect these events and contribute to inhibition of tumor growth?

As shown in Figure 5C, LNCaP tumors from SEMA3C ASO treated mice post castration shows reduced IHC staining with endothelial cell marker, CD31, as compared to tumors from Scr ASO treated controls suggesting that SEMA3C inhibition may in part contribute to tumor growth inhibition by inhibiting angiogenesis.

In fact, we have been actively exploring the pro-angiogenic effects of SEMA3C. We have found that full length SEMA3C is a potent pro-angiogenic factor while

inhibition of SEMA3C using SEMA3C shRNA inhibits angiogenesis. SEMA3C over-expression and shRNA-mediated silencing increased and decreased *in vitro* HUVEC cell tube formation, respectively. We postulate that the differences between our findings may be due to the SEMA3C constructs used in each of the studies. In our studies, we used uncleavable full length SEMA3C containing point mutations in both furin cleavage sites whereas Yang *et al.* 2015 used full length SEMA3C that can be proteolytically processed by furin.

SEMA3C induces HUVEC tube formation in MMRU melanoma cells. MMRU melanoma cells were transfected with either SEMA3C overexpression vector, shSEMA3C or vector control. The conditioned medium was collected and used for HUVEC tube formation assay. **(A)** SEMA3C overexpression and knockdown was confirmed by Western blot analysis. **(B)** SEMA3C overexpression induced while SEMA3C KD reduced the formation of tubular structures by HUVECs. **(C)** The number of tubes formed in each of the 5 randomly chosen fields. **, $P < 0.01$; ***, $P < 0.001$; Student's *t* test.

10. The authors recently published that SEMA3C expression in PC cells is regulated by androgens through GATA2 whereas FOXA1 inhibited SEMA3C expression. Do GATA2 and FOXA1 play a role in CRPC? Is their expression co-regulated with that of SEMA3C in CRPC specimens, or is there a different mode of SEMA3C regulation therein?

GATA2 and FOXA1 are coregulators of AR and are negatively prognostic for PCa. GATA2 expression correlates with tumour stage, predicts relapse and metastasis, and confers resistance to castration and chemotherapy in PCa¹. Likewise, FOXA1 is one of the most frequently mutated genes in CRPC^{2,3}. Hyperactivity by

coactivators of AR are thought to be a mechanism of CRPC. Thus, aberrant GATA2 and FOXA1 activity in CRPC may drive PCa progression. In RNA-Seq specimens of metastatic CRPC assembled by Robinson *et al*³, SEMA3C mRNA levels correlate with those of GATA2 and FOXA1 with Spearman correlations of 0.254 and 0.140, respectively. Rising GATA2 and FOXA1 levels may therefore be responsible for elevated SEMA3C levels in the CRPC landscape. However, we predict that a multitude of mechanisms exist independent of AR, GATA2, and FOXA1 which lead to the upregulation of SEMA3C in cancer such as stress-response and adaptive pathways.

- 1 Rodriguez-Bravo, V. *et al.* The role of GATA2 in lethal prostate cancer aggressiveness. *Nature reviews. Urology* **14**, 38-48, doi:10.1038/nrurol.2016.225 (2017).
- 2 Grasso, C. S. *et al.* The mutational landscape of lethal castration-resistant prostate cancer. *Nature* **487**, 239-243, doi:10.1038/nature11125 (2012).
- 3 Robinson, D. *et al.* Integrative Clinical Genomics of Advanced Prostate Cancer (vol 161, pg 1215, 2015). *Cell* **162**, 454-454, doi:10.1016/j.cell.2015.06.053 (2015).

11. There are a few important reports about the biological function and mechanism of action SEMA3C that are relevant to the current paper (e.g. effects on growth factor signaling pathways and signaling through Plexin D1 etc.) which were neither cited nor discussed by the authors. For example:

Yang et al. 2015 EMBO Mol Med. Semaphorin-3C signals through Neuropilin-1 and PlexinD1 receptors to inhibit pathological angiogenesis.

Plein et al. 2015 J Clin Invest. Neural crest-derived SEMA3C activates endothelial NRP1 for cardiac outflow tract septation.

Mumblat et al. 2015 Cancer Res. Full-Length Semaphorin-3C Is an Inhibitor of Tumor Lymphangiogenesis and Metastasis.

Toledano et al. 2016 Plos One. A Sema3C Mutant Resistant to Cleavage by Furin (FR-Sema3C) Inhibits Choroidal Neovascularization.

We thank the reviewer for his suggestion and have added a paragraph in the discussion relating to these reports.

Referee #3 (Remarks):

This manuscript investigates the roles of growth factor receptor tyrosine kinase (RTK) pathway activation in prostate cancer growth and anti-androgen treatment resistance. The authors found that SEMA3C drives activation of RTKs (EGFR, ErbB2, and MET) in a cognate ligand-independent manner via Plexin B1. Inhibition of SEMA3C-Plexin B1-RTK pathway through SEMA3C antisense oligonucleotides (SEMA3C ASO) or a Plexin B1 decoy protein (B1SP) decreases growth of castration-resistant prostate cancer (CRPC) and/or enzalutamide-resistant prostate cancer. Overall this is a very interesting and worthy area of research. The translational impact of this study is potentially high. The experiments are well conducted, and the results are clear and well presented.

Specific comments: While SEMA3C ASO or B1SP significantly inhibits growth of LNCaP-based castration-resistant/enzalutamide-resistant tumors in vivo (Figures 5,

6 and 8), it had moderate effect on growth of C4-2 CRPC cells in vivo (Figure S5H) (In fact C-4 is also a LNCaP-derived CRPC cell line). Will the in vivo growth of non-LNCaP derived CRPC cells (e.g. 22Rv1 cells) be affected by SEMA3C ASO or B1SP treatment?

We appreciate the referee's concern, also brought up by Referee 2, that the tumor volume of C4-2 xenografts shown in Figure S5H reached statistical significance at only the experimental end-point. The experiment shown, was done by injecting 2 million cells per site and two sites per mouse. C4-2 cells grow quite aggressively and because of the tumor burden the experiment had to be terminated before we could reach statistical significance beyond the endpoint as shown. To address this concern we repeated the C4-2 Xenograft experiment using 1 million cells/site and only one injection site per mouse using 5, control (PBS) and 7, B1SP-treated mice for treatment. We found that there was a better correlation between tumor volume and serum PSA levels when we used only one injection site per mouse. B1SP-mediated inhibition of C4-2 xenograft tumor volume achieved significance at week 6 following castration. This new data more clearly demonstrates tumour growth inhibition using B1SP (shown below) replaces the original data that now appears in Figure S6D and E.

Fig. S6 **(D)** Tumor volume (mm³) and **(E)** PSA (ng/ml) of from athymic *nu*^{-/-} mice bearing C42 Tumors treated with either PBS (n=5) or B1SP (n=7) post castration over a period of 10 weeks. Data are representative of Mean +/- SEM, **p*<0.05

2nd Editorial Decision

03 November 2017

Thank you for the submission of your revised manuscript to EMBO Molecular Medicine. We have now received the enclosed reports from the referees that were asked to re-assess it. As you will see the reviewers are now globally supportive and I am pleased to inform you that we will be able to accept your manuscript pending the following final amendments:

1) Please address the comments of referee 2. For point 1, we would recommend providing source data (see below, my point 4). Regarding point 3. Please make sure to introduce caution when

discussing figure 6A but we would like to encourage you to keep it as a main figure. Regarding point 4, please do let me know if you wish to follow the reviewer's suggestion or not and justify as appropriate.

***** Reviewer's comments *****

Referee #1 (Comments on Novelty/Model System for Author):

The model systems used are appropriate.

Referee #1 (Remarks for Author):

The authors have adequately addressed the concerns I raised previously.

Referee #2 (Remarks for Author):

1. For Figure S6B, control for Tumor 3 should be shown and the samples run on the same gel for facilitating comparison and interpretation of the data.
2. The data on different SEMA3C products in the different prostate cancer cells lines (that the authors have) should be shown and discussed in the paper.
3. Fig. 6A. The authors are not able to show additional tumor data. Given the natural heterogeneity in patient samples, these very low numbers need to be treated as preliminary and should be labeled and discussed as such, and perhaps moved to Supplementary figures.
4. Given that authors have data implicating SEMA3C in angiogenesis in this system (not shown in the paper, provided for the reviewers), consistent with previous publications, it brings up the possibility that the observed effects on tumor growth in vivo, at least in part, are mediated through this pathway, rather than through receptor tyrosine kinase signaling which is the main thrust of the paper. These data should be shown and this point elaborated upon in the Discussion.

Referee #3 (Comments on Novelty/Model System for Author):

The authors have successfully addressed my concerns by re-performing xenograft studies. The manuscript is suitable for publication.

2nd Revision - authors' response

22 November 2017

Mice bearing single LNCaP xenograft tumors were treated with control (PBS) or B1SP. A single control mouse and two B1SP mice were treated on the first occasion. We repeated the experiment using a single control mouse and a single B1SP-treated mouse. We have provided source blots for the data in Figures S6B in the Appendix.

2. The data on different SEMA3C products in the different prostate cancer cells lines (that the authors have) should be shown and discussed in the paper.
Given that the expression of SEMA3C in immortal benign prostate epithelial lines RWPE-1 and BPH-1 was very low, we moved this figure to Supplemental Figure 1B and replaced Figure 1C with the data showing the various SEMA3C products in the different prostate cancer cell lines. We have discussed the presence of the various SEMA3C secreted products in the results section.
3. Fig. 6A. The authors are not able to show additional tumor data. Given the natural heterogeneity in patient samples, these very low numbers need to be treated as preliminary and should be labeled and discussed as such, and perhaps moved to Supplementary figures.

We have changed the results section regarding the data presented in Figure 6A to reflect a more cautious preliminary interpretation of the presented data.

4. Given that authors have data implicating SEMA3C in angiogenesis in this system (not shown in the paper, provided for the reviewers), consistent with previous publications, it brings up the possibility that the observed effects on tumor growth *in vivo*, at least in part, are mediated through this pathway, rather than through receptor tyrosine kinase signaling which is the main thrust of the paper. These data should be shown and this point elaborated upon in the Discussion. It is likely that angiogenesis has a role to play in the effects of SEMA3C-driven prostate cancer. This is an area that we are actively pursuing and is beyond the scope of this report.

Corresponding Author Name: Christopher J. Ong

Manuscript Number: EMM-2017-07689